# RPS: Information Elicitation with Reinforcement Prompt Selection

## Abstract

Large language models (LLMs) have shown remarkable capabilities in dialogue generation and reasoning, yet their effectiveness in eliciting user-known but concealed information in open-ended conversations remains limited. In many interactive AI applications, such as personal assistants, tutoring systems, and legal or clinical support, users often withhold sensitive or uncertain information due to privacy concerns, ambiguity, or social hesitation. This makes it challenging for LLMs to gather complete and contextually relevant inputs. In this work, we define the problem of information elicitation in open-ended dialogue settings and propose Reinforcement Prompt Selection (RPS), a lightweight reinforcement learning framework that formulates prompt selection as a sequential decision-making problem. To analyze this problem in a controlled setting, we design a synthetic experiment, where a reinforcement learning agent outperforms a random query baseline, illustrating the potential of policy-based approaches for adaptive information elicitation. Building on this insight, RPS learns a policy over a pool of prompts to adaptively elicit concealed or incompletely expressed information from users through dialogue. We also introduce IELegal, a new benchmark dataset constructed from real legal case documents, which simulates dialogue-based information elicitation tasks aimed at uncovering case-relevant facts. In this setting, RPS outperforms static prompt baselines, demonstrating the effectiveness of adaptive prompt selection for eliciting critical information in LLM-driven dialogue systems.

## 1 Introduction

Large language models (LLMs) have demonstrated impressive capabilities in dialogue generation, reasoning, and information synthesis. Despite these strengths, their ability to effectively elicit concealed or uncertain information from users in open-ended conversations remains a significant challenge. In real-world domains such as health interviews (Li et al., 2024c; Qiu et al., 2024; Wang et al., 2025b), financial advising (Yu et al., 2024; Li et al., 2024b), and legal consultation (Cui et al., 2023; Li et al., 2024a; Sun et al., 2024), users often hesitate to disclose sensitive or uncertain information due to privacy concerns, self-presentation goals, or contextual ambiguity (Wang & Ziano, 2025). As a result, interactions become less informative and effective, often leading to suboptimal recommendations, missed diagnoses, or incomplete legal assessments. Making matters worse, such information is inherently difficult to annotate or collect, since it is purposefully withheld by users. To address this, a key step toward building more robust and context-aware conversational agents is the design of systems that can adaptively elicit concealed information without relying on explicit supervision or handcrafted rules.

With the advent of instruction-tuned LLMs, there is increasing interest in leveraging prompt engineering to guide these models toward more effective information-seeking behavior (Llanes-Jurado et al., 2024; Chen et al., 2024; Pang et al., 2024; Wang et al., 2025a). However, crafting optimal prompts remains labor-intensive and highly context-dependent. Moreover, without a principled formulation of the information elicitation problem, existing methods lack mechanisms to learn or adapt prompt strategies over time. Recent approaches to automated prompt search, such as RLPrompt (Deng et al., 2022) and GRIPS (Prasad et al., 2022), highlight the potential of reinforcement and gradient-free optimization respectively. Yet, both face practical limitations: RLPrompt introduces significant computational overhead due to token-level sequential generation and reward

evaluation. In addition to being inefficient, token-level generation can lack prompt diversity and makes it difficult to integrate domain-specific knowledge. GRIPS, though edit-based and gradient-free, lacks reinforcement-driven adaptation and requires frequent reward lookups during inference. These shortcomings motivate our proposed approach: a lightweight, reinforcement-based prompt selection framework that adaptively elicits concealed information with high efficiency, ensures prompt diversity, and facilitates integration of domain-specific knowledge, making it suitable for real-time open-ended dialogue.

In this paper, we define the problem of Information Elicitation (IE) in open-ended dialogue, where a language model must adaptively elicit information that users know but may conceal or only partially express during interaction. To analyze this problem in a controlled setting, we design a synthetic experiment using a Gaussian Mixture Model environment that highlights the core challenges of adaptive information elicitation. We then propose Reinforcement Prompt Selection (RPS), a lightweight reinforcement learning framework that formulates prompt selection as a sequential decision-making task. Unlike static or handcrafted approaches, RPS learns a policy over a pool of prompts and adapts its strategy based on user feedback to uncover information users may initially withhold. This lightweight and domain-agnostic approach enables scalable deployment in real-world applications. By adaptively selecting prompts from a predefined pool, RPS reduces reliance on static prompts, handcrafted rules, and costly token-level generation, while also promoting prompt diversity and supporting the integration of domain-specific knowledge. The key contributions of this paper are summarized as follows:

- We define the problem of **Information Elicitation (IE)** in open-ended dialogue, where a language model must uncover user-known but initially concealed information. To address this, we propose **Reinforcement Prompt Selection (RPS)**, a lightweight reinforcement learning framework that formulates prompt selection as a sequential decision-making task. RPS adaptively selects prompts from a predefined pool to promote user disclosure during conversation.

- We introduce **IELegal**, a benchmark dataset constructed from real legal case documents. It simulates an interactive dialogue in which the user represents the subject of a legal case, and the LLM must elicit key factual information. IELegal is designed to reflect the challenge of information elicitation in settings where users may deliberately withhold or obscure sensitive facts, making it a suitable testbed for evaluating models in complex and realistic legal consultation scenarios.

- We conduct experiments in both synthetic and real-world settings. In a controlled Gaussian Mixture Model environment, an RL agent outperforms a random query baseline, validating the challenge of adaptive information elicitation. In IELegal, RPS significantly outperforms static prompt baselines, demonstrating its effectiveness in uncovering relevant and potentially concealed information.

## 2 RELATED WORKS

**Information Elicitation.** Early works on information elicitation are largely rule-based, relying on predefined dialogue triggers or policies (Leal & Pearl, 2007; Tian et al., 2020). Leal & Pearl (2007) model elicitation as an interactive dialogue for decision tree construction, treating it as a heuristic search process. Tian et al. (2020) explore multi-round dialogue for eliciting user requirements, using knowledge graphs and granular computing to reduce conversational burden.

Learning-based approaches use deep neural networks to model and optimize information elicitation (Radlinski et al., 2019; Zuo et al., 2022; Kwan et al., 2023; Kostric et al., 2024). Radlinski et al. (2019) propose coached conversational preference elicitation, which guides users in expressing structured preferences through dialogue. Zuo et al. (2022) introduce hierarchical bandit algorithms that balance between asking about categories and recommending items, improving elicitation efficiency by modeling category and item-level rewards. Kwan et al. (2023) explore reinforcement learning methods for training dialogue systems to gather information through natural user interaction. Kostric et al. (2024) propose a novel strategy for conversational recommendation by asking usage-oriented questions rather than directly requesting item attributes, addressing the challenge that users may lack domain knowledge.

Recent advances in Large Language Models (LLMs) have enabled more powerful strategies for information elicitation (Llanes-Jurado et al., 2024; Pang et al., 2024; Gan et al., 2024; Wang et al., 2025a). Llanes-Jurado et al. (2024) introduce a virtual human system that combines LLM-driven semantic parsing, context-aware dialogue management, and adaptive question generation to extract user information in real-time interactions. Pang et al. (2024) propose a framework that empowers LLMs to actively reduce uncertainty through strategic questioning. Gan et al. (2024) present a benchmark designed to evaluate the ability of seeker agents to resolve uncertainty through clarification and information seeking questions. Wang et al. (2025a) introduce a meta-training approach that models uncertainty in information elicitation by simulating future observations.

**Discrete Prompt Optimization.** Prompt generation and selection play a critical role in effective information elicitation (Deng et al., 2022; Prasad et al., 2022; Zhang et al., 2022). Deng et al. (2022) propose a reinforcement learning approach that eliminates the dependence on gradient access or manual engineering by training a parameter-efficient policy network to generate discrete prompts via sequential token-wise selections. Prasad et al. (2022) introduce a gradient-free, edit-based method that improves instructional prompts through phrase-level operations such as delete, swap, paraphrase, and add, making it suitable for API-based language models. Zhang et al. (2022) present a test-time prompt editing framework that uses reinforcement learning to adapt and refine prompts for individual queries, supporting interpretable, query-specific prompt optimization across diverse tasks.

While prior work has advanced information elicitation through rule-based policies, learning-based methods, and large language models (LLMs), existing approaches often assume cooperative users, rely on static prompts or domain-specific heuristics, and lack adaptability in open-ended settings. In contrast, our method formulates prompt selection from a predefined pool as a reinforcement learning problem. This approach supports the use of diverse and even adversarial prompts designed to elicit information users may withhold. Moreover, it promotes prompt diversity, supports domain-specific integration, and reduces computational overhead compared to token-level generation. As a result, the approach offers a scalable and adaptive solution for eliciting user-known but potentially concealed information, particularly in real-time dialogue within sensitive and information-rich domains such as legal or clinical settings.

## 3 METHOD

### 3.1 INFORMATION ELICITATION

**Information.** We examine a time-slot framework $\mathcal{T} = \{1, 2, \ldots, T\}$ in which a user $u$ engages in open-ended dialogue with a language model $m$ for consultation or assistance (e.g., legal advice, health interviews). The user possesses a set of factual information $\mathcal{I}$ relevant to the dialogue, such as case details, symptoms, or personal circumstances. However, the user may choose to withhold some of this information. The goal of the model $m$ is to elicit as much factual information as possible through conversation, a process that we refer to as *Information Elicitation (IE)*. To formalize this setting, we assume that the user $u$ holds a complete set of information $\mathcal{I}$, while the model aims to recover relevant elements of $\mathcal{I}$ through a conversational interaction.

We categorize the user's information $\mathcal{I}$ into three types $\mathcal{C} = \{\mathcal{I}^+, \mathcal{I}^0, \mathcal{I}^-\}$ as follows:

- $\mathcal{I}^+$ **(Positive Information):** Information that reflects favorably on the user or supports their goals. Users are typically willing to share this voluntarily.
- $\mathcal{I}^0$ **(Neutral Information):** Information that neither benefits nor harms the user. Disclosure depends on context or prompting.
- $\mathcal{I}^-$ **(Negative Information):** Information that may be detrimental to the user's image, legal standing, or well-being. Users are generally reluctant to disclose this unless directly prompted.

Although the user holds the complete information set $\mathcal{I}$, their disclosure during interaction is often selective, influenced by both social and psychological factors. Our categorization of information types is motivated by insights from social psychology. Specifically, when prompted about $\mathcal{I}^+$, users often engage in self-enhancing behaviors driven by self-presentation theory (Goffman, 2023; Baumeister, 1982; Leary & Kowalski, 1990), readily revealing details that reflect positively on themselves. In contrast, users are inclined to withhold $\mathcal{I}^-$ due to factors such as cognitive dissonance,

impression management, or self-protection (Festinger, 1962; Leary, 2000). These behavioral tendencies present key challenges for models attempting to elicit sensitive or concealed information.

We represent the interaction between the model $m$ and the user $u$ as a sequence of dialogue rounds:

$$\mathcal{D} = \{\mathcal{D}_1, \mathcal{D}_2, \ldots, \mathcal{D}_T\}$$

where each round $\mathcal{D}_t = (q_t, v_t)$ consists of a model query $q_t$ and the corresponding user answer $v_t$.

The model's objective in the dialogue is to recover the complete information set $\mathcal{I}$ through a sequence of queries, including elements from the negative information $\mathcal{I}^-$ that the user may initially withhold. Note that the negative information $\mathcal{I}^-$ will be disclosed if the model's query is contextually appropriate and specific enough to elicit a relevant response.

**Query.** At each dialogue round $t$, the model $m$ formulates a question $q_t$ based on its currently known information $\hat{\mathcal{I}}_{t-1}$ and the previous dialogue history $\mathcal{D}_{1:t-1}$. This query generation process is defined as:

$$q_t = Q(\hat{\mathcal{I}}_{t-1}, \mathcal{D}_{1:t-1}), \tag{1}$$

where $Q(\cdot)$ denotes the model's query-generation function.

Upon receiving the query $q_t$, the user $u$ interprets it and identifies a target $x_t$ that they believe the question is referring to. This interpretation is shaped by the user's perspective, dialogue context, and subjective understanding of the query. We represent this process as:

$$x_t = f_u(q_t), \quad x_t \sim \mathcal{P}_u, \tag{2}$$

where $f_u(\cdot)$ denotes the user's interpretation function, and $\mathcal{P}_u$ is a probability distribution over possible interpretations of the query, reflecting how the user might understand or frame its intended meaning. Due to inherent cognitive and social biases, the user may respond selectively based on how they perceive the nature of $x_t$. Specifically, they are more likely to respond when $x_t$ aligns with positive or neutral content, and more likely to ignore or deflect when $x_t$ is perceived as sensitive or negative.

**Answer.** After interpreting the query $q_t$ and identifying a target interpretation $x_t \sim \mathcal{P}_u$, the user $u$ formulates a response $v_t$ based on the content of $x_t$, their internal information set $\mathcal{I}$, and the prior dialogue history $\mathcal{D}_{1:t-1}$. This user's response generation process is defined as:

$$v_t = V(x_t, \mathcal{I}, \mathcal{D}_{1:t-1}), \tag{3}$$

where $V(\cdot)$ denotes the user's answering function at round $t$. After receiving the user's response $v_t$, the model $m$ applies an information extraction function $f_m$ to identify any new factual content. We denote the extracted information as:

$$i_t = f_m(v_t), \tag{4}$$

where $i_t$ represents the newly identified facts in the current round. The model's known information is then updated by appending $i_t$ to the previous information set:

$$\hat{\mathcal{I}}_t = \hat{\mathcal{I}}_{t-1} \cup i_t. \tag{5}$$

### 3.2 REINFORCEMENT PROMPT SELECTION

In Figure 1, we introduce Reinforcement Prompt Selection (RPS), a framework that formulates information elicitation as a sequential decision-making problem. At each round of dialogue, the language model adaptively selects a prompt with the goal of maximizing cumulative improvements to its estimate of the user's hidden information $\hat{\mathcal{I}}$. By receiving feedback through user responses and updating its information state accordingly, the model learns a prompt selection policy that elicits information more effectively over time. RPS is trained using reinforcement learning, with a normalized reward function designed to help stabilize optimization and prevent performance degradation as the information gap narrows.

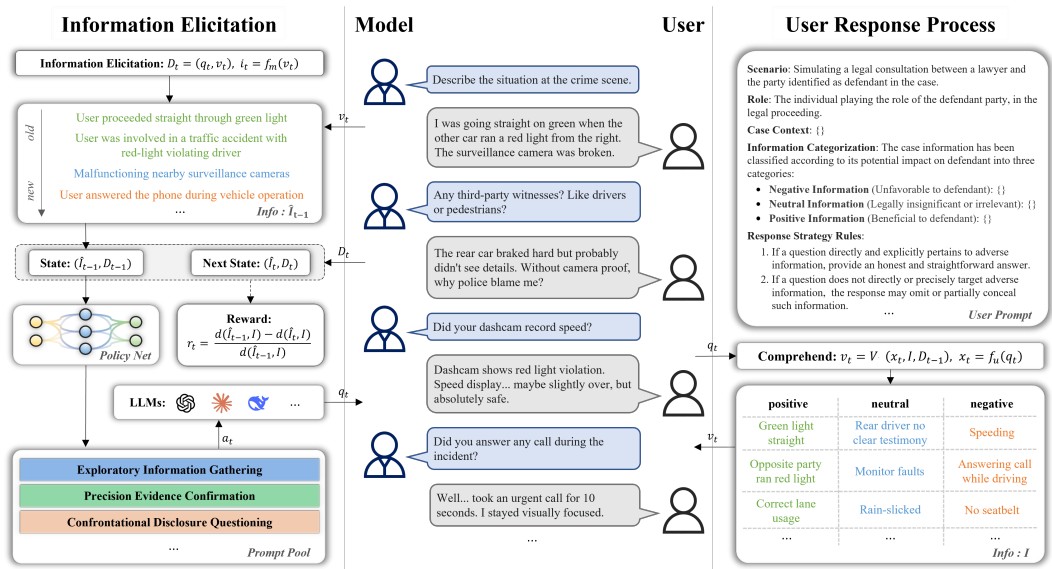

Figure 1: RPS uses reinforcement learning to adaptively select policy prompts that elicit users' concealed information $\mathcal{I}^-$ through multi-turn dialogue. At each turn $t$, the model updates its information estimate $\hat{\mathcal{I}}_{t-1}$, computes a reward $r_t$ based on information gain, and trains a policy $\pi_\theta$ to optimize prompt selection from a predefined pool.

**State.** At each round $t$, the model observes a state $s_t$ consisting of its current known information $\hat{\mathcal{I}}_{t-1}$ and the dialogue history $\mathcal{D}_{1:t-1}$. Formally,

$$s_t = (\hat{\mathcal{I}}_{t-1}, \mathcal{D}_{1:t-1}),$$

where $\hat{\mathcal{I}}_{t-1}$ is the accumulated information extracted from all previous user responses, as defined in Eq. equation 5. The state summarizes what the model has learned so far and guides the selection of the next prompt for generating new query.

**Action.** At each round $t$, the model selects a prompt $a_t \in \mathcal{A}$, where $\mathcal{A}$ is a predefined pool of prompts that defines the action space. Each prompt in $\mathcal{A}$ corresponds to a distinct information elicitation strategy (e.g., direct, precise, adversarial). The policy $\pi_\theta(a_t \mid s_t)$ maps the current state $s_t$ to a probability distribution over these prompts. Once a prompt $a_t$ is selected, it is combined with the model's current known information $\hat{\mathcal{I}}_{t-1}$ and dialogue history $\mathcal{D}_{1:t-1}$ to construct a final input for the language model. The language model then generates the next query $q_t$ using a query generation function:

$$q_t = \hat{Q}(a_t, \hat{\mathcal{I}}_{t-1}, \mathcal{D}_{1:t-1}),$$

where $\hat{Q}(\cdot)$ denotes the query generation process executed by the LLM, conditioned on the selected prompt and contextual inputs.

**Reward.** The reward $r_t$ is defined as a normalized information gain, measuring how much the estimated information $\hat{\mathcal{I}}_t$ moves closer to the ground-truth user information $\mathcal{I}$ relative to the previous round. Let $d(\cdot, \cdot)$ denote a distance metric between the estimated and true information representations. The reward is computed as:

$$r_t = \frac{d(\hat{\mathcal{I}}_{t-1}, \mathcal{I}) - d(\hat{\mathcal{I}}_t, \mathcal{I})}{d(\hat{\mathcal{I}}_{t-1}, \mathcal{I})}, \tag{6}$$

which reflects the proportion of remaining distance closed in round $t$. This normalization plays a crucial role in stabilizing optimization across the dialogue. Specifically, it stabilizes reward signals when the information gap is large and prevents the reward from vanishing as the model's estimate

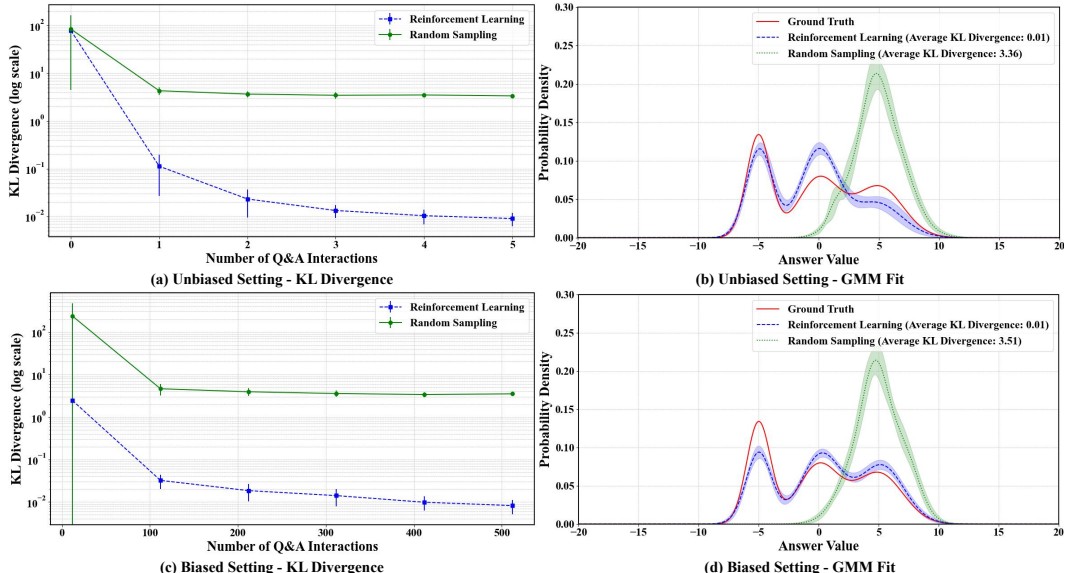

Figure 2: GMM-based simulation of adaptive information elicitation under unbiased and biased user disclosure. (a) and (c) report the KL divergence between the estimated and true information distributions over dialogue rounds for the unbiased and biased user settings. (b) and (d) visualize the corresponding GMM fits. Across both settings, the RL–based querying strategy consistently achieves lower KL divergence than the random querying baseline, and maintains a stable average divergence around 0.01 even when the user's disclosure distribution is biased, illustrating the robustness and effectiveness of learning-based strategies for information elicitation.

improves, ensuring continued learning even in later stages. In this way, the normalized reward facilitates effective and efficient elicitation of concealed information.

## 4 EXPERIMENTS

### 4.1 GAUSSIAN MIXTURE MODEL

**Experimental Setup.** We construct a simulated reinforcement learning environment based on a Gaussian Mixture Model (GMM) to emulate a multi-round dialogue between a model $m$ and a user $u$. This environment serves as a controlled testbed for evaluating information elicitation strategies under varying user disclosure behaviors.

The user's disclosure preferences are modeled as a probability distribution $\mathcal{P}_u(x)$ over a latent semantic space, represented by a GMM with components corresponding to three information categories $\mathcal{C} = \{+, 0, -\}$ (positive, neutral, negative):

$$\mathcal{P}_u(x) = \sum_{c \in \mathcal{C}} w_u^c \cdot \mathcal{N}(x \mid \mu_u^c, \Sigma_u^c), \tag{7}$$

$$\boldsymbol{w}_u = \{w_u^c\}_{c \in \mathcal{C}} \sim \text{Dirichlet}(\alpha_u^+, \alpha_u^0, \alpha_u^-), \tag{8}$$

where $\mathcal{N}(x \mid \mu_u^c, \Sigma_u^c)$ denotes the Gaussian for category $c$, and the mixture weights $\boldsymbol{w}_u$ reflect the user's disclosure bias.

At each round $t$, the model issues a query $q_t \in \mathbb{R}^d$, expressing its current information-seeking intent. The user interprets this query as an internal latent representation $x_t \sim \mathcal{P}_u$, following the process described in Eq. equation 2. The user then evaluates the posterior probability of each disclosure category $c \in \mathcal{C}$ given $x_t$, and selects the most likely category

$$\hat{c}_t = \arg\max_{c \in \mathcal{C}} \left\{ \frac{w_u^c \, \mathcal{N}\left(x_t \mid \mu_u^c, \Sigma_u^c\right)}{\sum_{c' \in \mathcal{C}} w_u^{c'} \, \mathcal{N}\left(x_t \mid \mu_u^{c'}, \Sigma_u^{c'}\right)} \right\}. \tag{9}$$

A feedback response $v_t$ is then sampled from the Gaussian associated with the selected category

$$v_t \sim \mathcal{N}(\mu_u^{\hat{c}_t}, \Sigma_u^{\hat{c}_t}). \tag{10}$$

After generating a response $v_t$, the model treats it as the extracted information for that round, denoted $i_t = v_t$. The sequence $\{i_1, i_2, \ldots, i_t\}$ forms the accumulated evidence from which the model estimates the user's underlying information $\hat{\mathcal{I}}_t$, modeled as a GMM with $|\mathcal{C}|$ components fit to the collected responses.

To evaluate information recovery, we assume the ground-truth information distribution $\mathcal{I}$ follows a GMM with fixed component parameters but uniform weights across all categories. This reflects the assumption that, in principle, positive, neutral, and negative information are equally likely to occur in the underlying semantic space, independent of user-specific disclosure preferences.

We quantify performance using the average of component-wise KL divergences between the model's estimate $\hat{\mathcal{I}}_t$ and the true distribution $\mathcal{I}$. To resolve permutation ambiguity between mixture components, we apply the Hungarian algorithm to find the optimal matching $\sigma$ between the estimated and true components. The resulting distance function used in the reward computation (Eq. equation 6) is defined as

$$d(\hat{\mathcal{I}}_t, \mathcal{I}) = \frac{1}{|\mathcal{C}|} \sum_{c=1}^{|\mathcal{C}|} \mathrm{KL}\left( \hat{I}_t^{\sigma(c)} \parallel I^c \right). \tag{11}$$

This enables reinforcement learning based on progressive information gain.

**Experimental Results.** We use the GMM-based simulation to validate the core challenge of adaptive information elicitation and to illustrate the potential of reinforcement learning–based strategies in a controlled setting. We consider two user configurations: (1) a biased user, where the disclosure distribution $\mathcal{P}_u(x)$ assigns greater weight to certain information categories (e.g., favoring positive content) and deviates from the ground-truth distribution $\mathcal{I}$, which is defined as a uniformly weighted GMM over all categories; and (2) an unbiased user, where $\mathcal{P}_u(x) = \mathcal{I}$, modeling a fully cooperative case. In both scenarios, the model interacts over multiple dialogue rounds, accumulates extracted responses $\{i_1, i_2, \ldots, i_t\}$, and fits a GMM $\hat{\mathcal{I}}_t$ to approximate the true information distribution. Due to the continuous nature of the latent semantic space in this environment, the action space is defined over continuous query vectors rather than discrete prompt selections. More details are provided in Appendix B.1.

We compare the reinforcement learning–based method against a random querying baseline. Query effectiveness is measured using the KL divergence $d(\hat{\mathcal{I}}_t, \mathcal{I})$, which measures the distance between the estimated and true distributions. The results are averaged over 10 independent runs to ensure statistical reliability. As shown in Figure 2, the reinforcement learning method consistently achieves lower KL divergence across dialogue rounds compared to the random baseline. In both the unbiased and biased settings, the reinforcement learning method maintains a stable average KL divergence of 0.01, demonstrating its robustness to disclosure bias. In the biased setting, the random baseline deteriorates significantly (from 3.36 to 3.51), indicating the relative robustness of learning-based strategies under disclosure bias.

## 4.2 LLM-BASED DIALOGUE

**IELegal Dataset.** To evaluate information elicitation in a realistic domain setting, we construct *IELegal*, a legal dialogue benchmark based on criminal court records. IELegal comprises two subsets: *IELegal-base*, which includes 1,000 real-world cases with structured factual annotations, and *IELegal-augment*, an enriched version where sparse narratives are expanded with additional contextual detail.

The dataset is derived from the CAIL2018 corpus, a large-scale collection of Chinese criminal cases (Xiao et al., 2018). Each case includes a factual description field, from which we use a large

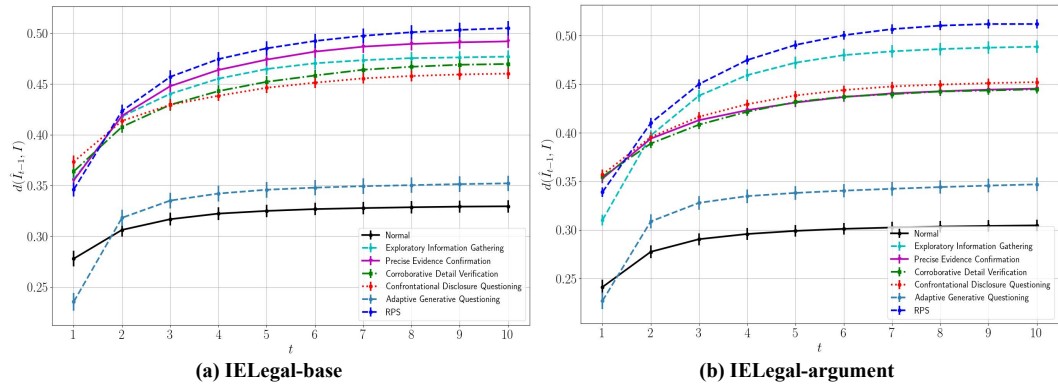

(a) IELegal-base  (b) IELegal-argument

Figure 3: Comparison of RPS (blue) against baselines on the (a) IELegal-base and (b) IELegal-augment datasets. The y-axis indicates the semantic similarity between the cumulative extracted information and the ground truth information in different rounds $t$. RPS consistently outperforms all baselines by the end of the dialogue.

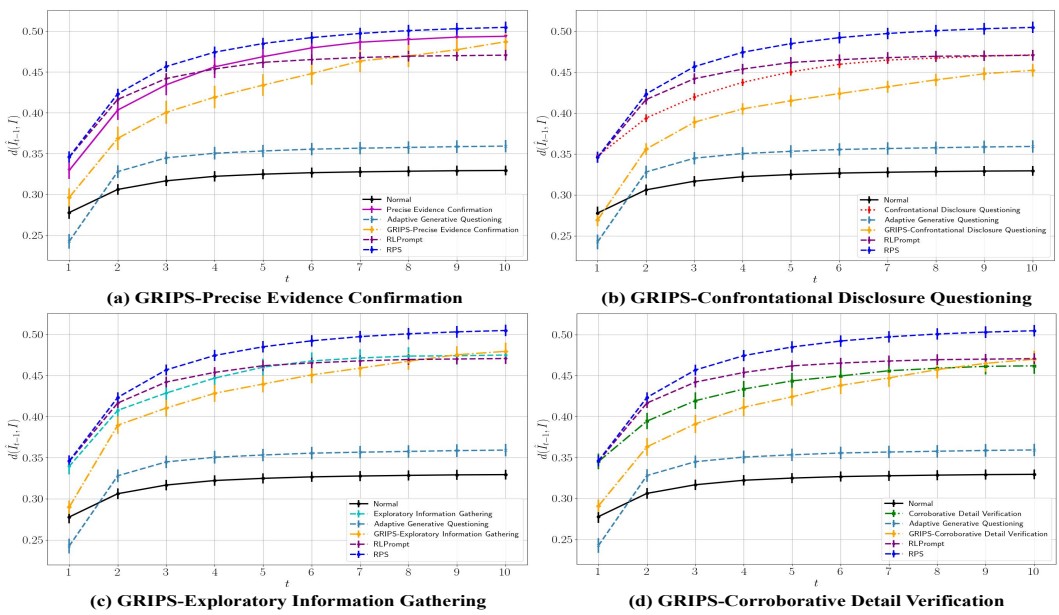

(a) GRIPS-Precise Evidence Confirmation  (b) GRIPS-Confrontational Disclosure Questioning

(c) GRIPS-Exploratory Information Gathering  (d) GRIPS-Corroborative Detail Verification

Figure 4: Comparison of RPS (blue), RLPrompt and the GRIPS variant based on four fixed prompt strategies on the IELegal-base dataset. The performance of both GRIPS and RLPrompt remains consistently inferior to that of RPS, and in several cases even falls below the results obtained using the original domain-specific prompts alone.

language model (i.e. GPT-4o-mini) to extract key factual elements such as time, location, participants, and actions. These structured elements serve as the reference ground truth for evaluating how much information a dialogue agent can recover through multi-round interactions.

To simulate user disclosure bias, we use the language model to categorize each extracted fact as positive, neutral, or negative based on its implications for the defendant. This enables dynamic user behavior in simulation, where unfavorable information may be withheld unless effectively elicited.

Because many legal cases contain limited detail, we create *IELegal-augment* by applying LLM-based augmentation to expand factual narratives with plausible contextual information. This results in a more challenging benchmark that better reflects real-world variation in case complexity. Addi-

tional details including dataset construction procedures, prompt designs, data examples, and dataset statistics are provided in Appendix D.

**Baselines.** We compare the proposed RPS framework with eight baselines, including four fixed prompt strategies, one default strategy without any prompting (**Normal**), one prompt strategy named "Adaptive Generative Questioning" produced by the large language model , GRIPS and RLPrompt. Each strategy guides the language model to play the role of a lawyer, adopting a distinct questioning style $a_t \in \mathcal{A}$ to elicit target information $\mathcal{I}$ from the user through dialogue. These strategies are grounded in established legal interviewing practices (Ying zai tan pan, 2010; Han & Peng, 2015; 2024). The four fixed prompt strategies and Adaptive Generative Questioning are as follows and more details can be found in Appendix C.2:

- **Exploratory Information Gathering** encourages open-ended dialogue to elicit broad and spontaneous user responses. By using prompts such as *"Could you describe in detail what happened on that day?"*, the model facilitates free narration, which can uncover latent facts and reconstruct the case context.

- **Precise Evidence Confirmation** targets the elicitation of specific factual details to support the verification of critical elements. This strategy uses closed or narrowly focused questions to clarify key aspects of the case. For example: *"Were you carrying anything in your hands at that moment?"* or *"Do you remember what time you arrived at the location?"* These prompts aim to reduce ambiguity and reinforce evidentiary consistency.

- **Corroborative Detail Verification** involves cross-validating information from multiple perspectives to ensure completeness and internal consistency. This strategy focuses on follow-up questions that clarify vague points, recover missing details, or identify contradictions. It examines facts through varied lenses, including individual behavior, environmental context, and supporting evidence. For example: *"Can you describe in detail the person's clothing and physical features at that moment?"*

- **Confrontational Disclosure Questioning** targets information that the user may deliberately withhold due to its sensitive or unfavorable nature. This strategy prompts users to elaborate on omitted details by drawing attention to gaps in their narrative. For example: *"You said you were at the scene to talk. Were there any physical actions or interactions that took place during that time?"*

- **Adaptive Generative Questioning** adopts a dynamic policy-generation mechanism. Prior to each conversational round, the system generates a lawyer-specific policy for the subsequent round of questioning with a large language model, based on the complete cumulative dialogue history of the case and a predefined set of prompts. This policy-generation process is recursive: as the dialogue progresses, the model continually performs conditional generation over the evolving dialogue state and the strategic prompts, iteratively producing new lawyer policies until the prescribed maximum number of turns for the case is reached.

- **GRIPS** performs rule-based prompt editing via simple operations on the instruction. We adapt the original GRIPS framework to the multi-turn dialogue setting as follows: We use each of the first four domain knowledge based strategies as an initial prompt for GRIPS. After each dialogue turn, GRIPS dynamically adjusts the strategy based on the dialogue history (e.g., by deleting, swapping, or adding segments), and the updated prompt is then used in the next turn. Further implementation details are provided in Appendix B.2.1.

- **RLPrompt** introduces a gradient-free RL framework that optimizes discrete prompt generation via sequential token selection, obviating the need for manual design. We adopt the complete RLPrompt framework for discrete prompt generation. To maintain a fair comparison, we substitute its original reward function with the one proposed in this paper. Further implementation details are provided in Appendix B.2.2.

**Experimental Setup.** To assess the effectiveness of dialogue-based IE, we design an automatic evaluation framework based on semantic similarity between the extracted information and the ground truth information across multiple rounds of conversation. We simulate the user using a language model prompted with background context. To model real-world disclosure bias, the prompt is designed to discourage the revelation of unfavorable (negative) information. This setup resembles the biased behavior in Eq. equation 9 of the GMM environment, where user responses are shaped by

category-dependent preferences. We use a LLMs to extract information $i_t$ from the user's response $v_t$, which is then added to the accumulated knowledge $\hat{\mathcal{I}}_t$. In the GMM setting, this corresponds to $i_t = v_t$.

To evaluate the performance of information elicitation, we measure the semantic similarity between the accumulated extracted information $\hat{\mathcal{I}}_t$ and the ground truth $\mathcal{I}$. We apply a language model to extract a structured set of factual elements from $\mathcal{I}$, including attributes such as time, location, participants, and events, resulting in a reference set $\mathcal{K} = \{\mathbf{k}_1, \ldots, \mathbf{k}_n\}$. At each round $t$, the distance is computed as:

$$d(\hat{\mathcal{I}}_t, \mathcal{I}) = \frac{1}{|\hat{\mathcal{I}}_t|} \sum_{j=1}^{|\hat{\mathcal{I}}_t|} \max_{k \in \mathcal{K}} \mathrm{sim}(i_t^{(j)}, k), \tag{12}$$

where $\mathrm{sim}(\cdot, \cdot)$ denotes cosine similarity between Sentence-BERT (Reimers & Gurevych, 2019) embeddings. In Eq. equation 12, we compute the maximum similarity between each extracted item $\mathbf{i}_t$ and the reference set $\mathcal{K}$ to capture its closest matching fact, as each response typically aligns with a single key element. The resulting distance metric serves as a continuous approximation of information alignment and parallels the KL-divergence distance used in the GMM setting (Eq. equation 11). We adopt Sentence-BERT for reward computation, as it offers both efficiency and stability, in contrast to LLM-based scoring, which is computationally expensive and exhibits unstable reward signals.

**Experimental Results.** Figure 3 compares the performance of RPS against six baselines on the *IELegal-base* and *IELegal-augment* datasets. The y-axis indicates the semantic similarity between the cumulative extracted information and the ground truth information in different rounds $t$. RPS consistently outperforms all baselines by the end of the dialogue, confirming its advantage in adaptively selecting effective prompt strategies. Among fixed strategies, "Normal" strategy performs worst due to the lack of prompt guidance. In *IELegal-augment*, RPS and the second-best strategy, "Exploratory Information Gathering", initially lag but surpass others in later rounds. This suggests that in richer factual contexts, overly specific prompts early in the dialogue may limit information coverage, whereas broader questioning strategies such as exploratory questioning may be more effective for information elicitation across multiple interactions. Notably, "Adaptive Generative Questioning" performs worse than several fixed strategies grounded in domain knowledge, suggesting that in the absence of sufficient domain knowledge, prompts generated by large language models may be limited in their effectiveness at eliciting critical information.

Figure 4 compares the performance of RPS, RLPrompt, and the GRIPS variant using the first four domain-informed strategies on the IELegal-base dataset. The results show that RPS consistently achieves the highest overall performance. Neither GRIPS nor RLPrompt surpasses RPS under any setting, and in most cases their performance is only comparable to that of the original fixed strategies. These findings suggest that rule-based prompt editing, whether based on simple instruction modifications as in GRIPS or reinforcement-based adjustments as in RLPrompt, has limited effectiveness in capturing and leveraging domain-specific knowledge in legal consultation and potentially in other specialized domains.

## 5 CONCLUSION

We address the problem of Information Elicitation (IE) in open-ended dialogue, where a language model aims to uncover information users may intentionally withhold. We propose Reinforcement Prompt Selection (RPS), a lightweight framework that formulates prompt selection as a sequential decision-making process. To support the evaluation, we introduce IELegal, a dialogue benchmark based on real legal cases, designed to reflect realistic user disclosure behavior. Experiments in both synthetic and real-world settings show that RPS significantly outperforms static prompting strategies in eliciting relevant and sensitive information. Despite these gains, several limitations remain. The current approach provides a theoretical foundation, but can be further enhanced with time-varying synthetic modeling, while real-world datasets could be extended beyond legal cases to domains such as healthcare. Future work includes developing more principled probabilistic models for IE, improving the quality and diversity of benchmark datasets, and extending RPS to other application domains such as healthcare and customer service.

ETHICS STATEMENT

The authors of this work have read and commit to adhering to the Code of Ethics. Our research studies the problem of adaptive information elicitation in dialogue using large language models (LLMs), introducing a reinforcement-based prompt selection framework. All experiments were conducted in controlled settings with either synthetic data or our newly created IELegal benchmark. IELegal was constructed from publicly available legal case documents, and all data have been thoroughly anonymized to remove personal identifiers and sensitive attributes. The dataset does not include private or confidential information, nor does it involve human subjects. The work does not deploy the proposed methods in real-world legal or clinical contexts, and therefore does not pose direct social or individual risks. Future applications of this research in sensitive domains such as law or healthcare should carefully address privacy, fairness, and accountability considerations to ensure ethical and responsible use. For deployment in real-world decision-making scenarios, adopting a human-in-the-loop paradigm is essential. Specifically, domain experts, such as licensed attorneys or clinicians, must remain responsible for interpreting model outputs, overseeing the dialogue history, and rendering final decisions. The system is intended strictly as an assistive tool designed to augment rather than replace professional expertise. Furthermore, implementation must prioritize user transparency and adhere to robust informed consent protocols to mitigate risks of misuse or unintended harm.

REPRODUCIBILITY STATEMENT

We provide the complete source code in the supplementary materials. Further details on the experimental setup, including hyperparameters, datasets, model architecture and prompt templates, are documented in the Appendix.

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

# Appendix

## A  DISCUSSION

### A.1  LIMITATIONS

Despite the progress made in this study, several limitations remain. First, the dataset used in our experiments is relatively small in scale, as both the IELegal-base and IELegal-augment subsets contain only 1,000 cases, resulting in limited coverage. Second, the dataset is restricted to the legal domain and does not include other potential application scenarios, such as medical consultations or psychological counseling. Finally, the Gaussian Mixture Model (GMM) experiment employed in this work does not incorporate temporal dynamics, leading to a gap between the simulated environment and real-world conversational settings.

### A.2  USE OF LARGE LANGUAGE MODELS

In this work, we used large language models (LLMs) to assist with manuscript editing. LLMs were used to help polish the language of the manuscript. This includes surface-level edits such as improving clarity, grammar, and conciseness of English expressions. All technical content, algorithmic designs, and empirical results were authored and validated by the authors. No part of the scientific contributions was generated by or delegated to an LLM.

## B  EXPERIMENTAL SETUP

### B.1  GAUSSIAN MIXTURE MODEL

This section details the experimental setup for our GMM-based dialogue experiment, including environment design, model configuration, and training procedures.

We implement a Gaussian-based reinforcement learning environment using the OpenAI Gym interface to simulate a dialogue scenario between a model $m$ (e.g., a lawyer) and a user $u$ (e.g., a client) (Brockman et al., 2016). The user's ground truth information $\mathcal{I}$ is represented by a Gaussian Mixture Model (GMM) with three components. Both the agent's actions and the user's observations are modeled as continuous scalar values.

Policy learning is conducted using the Deep Deterministic Policy Gradient (DDPG) algorithm (Lillicrap et al., 2015), with actor and critic networks. Both networks consist of fully connected layers with 128 hidden units and ReLU activations. The Adam optimizer is used (Kingma, 2014), with learning rates of $3 \times 10^{-3}$ for the actor and $3 \times 10^{-2}$ for the critic. A discount factor of $\gamma = 0.98$ is applied, and target networks are updated using a soft update rate of $\tau = 0.005$. Gaussian noise with standard deviation $\sigma = 0.01$ is added to the action outputs to encourage exploration. Training uses an experience replay buffer of 30,000 transitions, with learning initiated after an initial collection of 3,000 samples. Mini-batches of size 64 are sampled during training. The model is trained over 1,000 episodes of 500 steps each. All experiments are compatible with CPU-only execution.

At the beginning of each episode, the means of the GMM components are sampled from the interval $[-10, 10]$, with pairwise distances drawn uniformly from the range $[4, 5]$ to ensure spatial separation. The means are then sorted in descending order and assigned to the positive, neutral, and negative latent categories, respectively. The covariance matrices are diagonal, with variances sampled uniformly from $[1, 8]$, then sorted in descending order so that the positive component has the highest variance, followed by the neutral and negative components. This design reflects the assumption that positively valenced information exhibits greater variability in user disclosure behavior. Mixture weights are sampled from a Dirichlet distribution with concentration parameters $\alpha_u^+ = 3$, $\alpha_u^0 = 2$, and $\alpha_u^- = 1$, reflecting a prior tendency toward positive disclosures. Three GMM instances are maintained during each episode, including a ground truth model, a Dirichlet-weighted user model, and an adaptive model updated through interaction.

## B.2 LLM-BASED DIALOGUE

We construct an LLM-based reinforcement learning environment to evaluate the Reinforcement Prompt Search (RPS) algorithm. The environment simulates a dialogue scenario between the model agent $m$ (e.g., a lawyer) and the user $u$ (e.g., a suspect), where the user's information state $\mathcal{I}$ is drawn from the real-world dataset *IELegal*. The model $m$ selects actions from a prompt pool and dynamically adapts its dialogue strategy based on the user's responses and the algorithmic reward.

Policy learning is implemented using Deep Q-Network (DQN) (Mnih et al., 2013). The Q-network comprises four fully connected layers. The first three layers each have 128 hidden units. A ReLU activation is applied after the second layer, and a skip connection is introduced between the output of the first layer and the input of the third layer. A residual addition is then followed by layer normalization to stabilize training.

The model is trained using the following hyperparameters: a learning rate of $1 \times 10^{-2}$, a discount factor of $\gamma = 0.98$, and a target network update frequency of every 10 episodes. Experience replay is employed with a buffer size of 2,000, a minimum replay size of 500 before training begins, and a batch size of 64. Training runs for a total of 500 episodes. Each training episode consists of 10 dialogue rounds. At the beginning of an episode, the model $m$ randomly selects an initial prompt. In subsequent rounds, the model selects the dialogue strategy based on the state. We use Sentence-BERT to evaluate the semantic similarity between the information disclosed in each round and the ground truth (Reimers & Gurevych, 2019). The performance is reported with a 95% confidence interval.

### B.2.1 IMPLEMENTATION OF GRIPS BASELINE

**1. Policy Decomposition and Atomic Operations.**

The foundation of GRIPS is the editing and searching of the policy prompt.

- **Policy Decomposition:** We treat the lawyer's Policy Prompt as a sequence of semantic units. segment the policy text into Sentence-level Phrases based on punctuation marks (Chinese periods, question marks, exclamation points, and newlines). These phrases serve as the fundamental atomic units for GRIPS's editing operations.

- **Edit Operation Space:** We defined four gradient-free edit operators:
    - Delete: Randomly removes a phrase from the policy and places it in the "Deleted Pool."
    - Swap: Randomly exchanges the positions of two phrases to alter the logical flow of the instruction.
    - Paraphrase: Uses an LLM to semantically rephrase a selected phrase. We incorporate the current Dialogue History as context, ensuring the revised strategy is clearer and more actionable while maintaining its core objective.
    - Add: Randomly retrieves a phrase from the "Deleted Pool" and inserts it at an arbitrary position in the policy (triggered only when the pool is non-empty).

**2. Experimental Procedure.**

- **Initialization:** The dialog begins with the initial policy prompt corresponding to the fixed strategy.

- **Context-Aware Editing:** The GRIPS performs gradient-free edit operations (such as deletion, rephrasing, or swapping of semantic units) conditioned on the current dialog history to generate a set of candidate policy prompts.

- **Evaluation and Selection:** An independent LLM evaluator (serving as a proxy reward function) assesses the potential efficacy of each candidate prompt in eliciting critical new information in the subsequent turn. In our experiments, we use Qwen-32b as the LLM evaluator. The prompt with the highest score is then greedily selected as the updated policy prompt.

**3. Prompt.**

**Context-Aware Editing Prompt.**

```
strategy

Scenario:  Simulating an initial consultation between a
lawyer and a client to gather factual information about a
legal case.
Role:  You are a strategy expert, specialized in refining and
optimizing questioning strategy.
Task:  Based on the existing dialogue history and the
strategy used in the previous turn, you are required to
refine and rewrite the questioning strategy in a professional
and clear manner, while keeping the core objective of
uncovering the client's true information as your primary
goal.
Dialogue history:  {}
Strategy:  {}
Notice:  Your response should contain only the revised
Strategy.
```

**Evaluation Prompt.**

```
strategy

Scenario:  Simulating an initial consultation between a
lawyer and a client to gather factual information about a
legal case.
Role:  You are a strategy evaluation expert, specialized in
assessing the effectiveness of questioning strategy.
Task:  Your task is to evaluate how effective this strategy
will be in helping the lawyer obtain key factual information
in the next turn, based on the dialogue history and the given
strategy prompt.  You should output a single numerical score
from 1 to 10.
Dialogue history:  {}
Strategy:  {}
```

### B.2.2 IMPLEMENTATION OF RLPROMPT BASELINE

**1. Problem Formalization.**

The discrete prompt optimization is formalized as a Markov Decision Process (MDP):

- **Objective.** The goal is to train a Policy Network $\pi_\theta$ to maximize the expected reward $R$, ensuring the generated fixed-length discrete prompt sequence $\mathbf{z} =$ maximizes the performance on open-ended dialogue.

- **Environment.** The dialogue logic in the legal field is treated as a black-box environment. Crucially, the policy network does not require access to internal gradients.

- **State ($s$).** The state is defined by the current dialogue context which includes the most recent QA turns, ensuring the policy can generate an input-specific prompt based on the current case status.

- **Action ($a$).** The action space is the sequential selection of a discrete token $z_t$ from the vocabulary at each time step $t$, forming the fixed-length prompt sequence $\mathbf{z}$ (e.g., $T = 5$ tokens).

**2. Policy Network.**

- **Frozen Policy LM.** The Policy Network is constructed upon a frozen, compact pre-trained LM (e.g., distilgpt2), which functions to encode the state $s$ and the partial prompt $z_{<t}$

into high-quality contextual embeddings. We use Qwen2.5-0.5B (Yang et al., 2024) as the Policy Network.

- **Trainable MLP.** The component containing all trainable parameters is a small MLP layer. This MLP processes the embeddings, converting them into "adapted embeddings" which are then used by the LM Head to predict the probability distribution for the next token $z_t$.

**3. Sequence Exploration and Log Likelihood Tracking.**

During the training phase, the policy network must actively explore the discrete prompt space and generate the necessary data for gradient calculation.

- **Exploration (Top-K Sampling).** The policy network uses Top-K sampling (e.g., $K = 256$) to probabilistically select the next token $z_t$.
- **Log Likelihood Tracking.** For accurate policy gradient calculation, we defined a method cumulatively tracks the log likelihood of the entire sampled sequence $\mathbf{z}$: $\sum_{t=1}^{T} \log P(z_t|z_{<t})$. This resulting tensor is critical for the subsequent loss computation.

**4. Environment Interaction, Reward Calculation and Stabilization.**

- **Environment Interaction.** Generate a batch of candidate lawyer prompts ($\mathbf{Z}(s)$) for the same conversation state $s$, and sequentially conduct multiple rounds of dialogue.
- **Reward Calculation.** Compute a scalar, piecewise reward $R$, which is a composite metric.
- **Reward Stabilization.** Apply input-specific Z-Score normalization. This technique standardizes the raw reward $R$ across the batch $\mathbf{Z}(s)$, thereby counteracting training bias caused by intrinsic difficulty variations across different cases (inputs $x$). The normalization ensures that the policy updates are driven by relative performance improvement.

**5. Policy Update.**

We implement the policy gradient objective, minimized using the formula:

$$\text{Loss} = -\mathbb{E}[A \cdot \log \pi_\theta(\mathbf{z}|s)]$$

The Normalized Rewards (from 4) function as the Advantage $A$, and the optimization updates the MLP parameters derived from the accumulated $\log \pi_\theta(\mathbf{z}|s)$.

## C    AGENT

This section presents the prompts used for both the user and the model, including the scenario description, role definitions, case context, information categorization, response strategy rules, and response limitations.

### C.1    ROLE

**User Prompt.**    The user prompt defines a legal consultation scenario, where the user plays the role of a client seeking legal advice. The prompt includes case-specific context and outlines response strategy rules indicating that the user is more inclined to disclose information that is favorable to their case, while withholding or downplaying unfavorable details. A constraint is also specified: each user response must not exceed 30 Chinese characters.

---

**Example Case**

```
Scenario:  Simulating a legal consultation between a lawyer
and the party identified as defendant in the case.
Role:  The individual playing the role of the defendant
party, in the legal proceeding.
Case Context:  {}
```

---

```
Information Categorization:  The case information has been
classified according to its potential impact on defendant
into three categories:
    • Negative Information (Unfavorable to defendant):  {}
    • Neutral Information (Legally insignificant or
      irrelevant):  {}
    • Positive Information (Beneficial to defendant):  {}
Response Strategy Rules:
    1. If a question directly and explicitly pertains
       to adverse information, provide an honest and
       straightforward answer.
    2. If a question does not directly or precisely target
       adverse information, the response may omit or partially
       conceal such information.
    3. All questions regarding favorable or neutral
       information must be answered truthfully and completely.
    4. For questions concerning aspects not mentioned in the
       case materials, respond with "I don't know."
Limitation:  Each response must be no longer than 30 Chinese
characters (or equivalently concise in another language), to
simulate realistic and focused user replies.
```

**Model Prompt.** The model prompt specifies that, within the context of a simulated initial consultation between a lawyer and a client, the model is required to assume the role of an experienced legal professional. It is expected to employ a structured interview strategy to elicit comprehensive case information from the user. This approach aims to mitigate common challenges such as information omissions or intentional concealment that may arise with non-professional users, thereby enhancing the accuracy of case analysis and the formulation of response strategies.

The prompt includes detailed elements such as scenario setting, role definition, task objectives, and specific interview goals. Additionally, a set of communication strategies is provided to standardize the formulation of questions during the interaction. A constraint is imposed such that each question must adhere strictly to the predefined communication strategy. Furthermore, each dialogue round is limited to a single question, and each response must not exceed 40 Chinese characters.

---

**Example Case**

```
Scenario:  Simulating an initial consultation between a
lawyer and a client to gather factual information about a
legal case.
Role:  You are an experienced lawyer, well-versed in various
communication techniques and capable of conducting structured
and strategic interviews.
Task:  Users often lack legal expertise and may inadvertently
omit or intentionally conceal critical information during
the consultation process.  These omissions may impair the
lawyer's ability to accurately assess the case and formulate
effective legal strategies.  To mitigate this risk, you are
tasked with employing structured communication strategies to
guide the user and obtain a comprehensive understanding of
the case.
Objectives:  Your questions should aim to elicit a complete
and detailed picture of the case, including the time,
location, persons involved, sequence of events, relevant
```

```
circumstances, tools used, available evidence, motivations,
intentions, and legal or factual consequences.
Communication Strategies:  Apply the following questioning
strategies throughout the conversation to elicit complete and
reliable information from the client:{}
Requirements:
   1. Maintain strict adherence to the specified
      communication strategies throughout the interaction.

   2. After receiving each client response, analyze the reply
      carefully and formulate a follow-up question according
      to the relevant strategy.

   3. Each question should be concise, with a maximum length
      of 40 Chinese characters (or equivalent in other
      languages).

   4. Proceed with one question at a time to ensure clarity
      and focus.
```

## C.2    POLICY

**Strategy 1: Exploratory Information Gathering.**    This strategy encourages open-ended dialogue to elicit broad and spontaneous user responses. By using prompts such as *"Could you describe in detail what happened on that day?"*, the model facilitates free narration, which can uncover latent facts and reconstruct the case context.

```
strategy

Core Objective:  To encourage clients to speak freely,
enabling comprehensive information gathering and the
discovery of potential details.
Key Points:
   1. Avoid Presupposed Answers:  Questions should be framed
      in a way that avoids imposing predetermined directions
      or assumptions.  This allows clients to narrate events
      in their own words and logical order, thereby revealing
      more authentic and complete information.

   2. Stimulate Expressiveness:  By utilizing open-ended
      questions such as "how" and "what" lawyers can
      prompt clients to elaborate on the background of the
      events and express their emotional responses.  This
      facilitates a deeper understanding of the case context.

   3. Observe Nonverbal Signals:  While clients are speaking,
      attention should be paid to their emotions, tone of
      voice, and body language, which may reveal hidden
      contradictions or critical information.

Example Questions:
   • "Could you describe in detail what happened on that
     day?"

   • "Are there any additional details regarding the dispute
     that you would like to add?"
```

**Strategy 2: Precise Evidence Confirmation.**    This strategy targets the elicitation of specific factual details to support the verification of critical elements. This strategy uses closed or narrowly focused questions to clarify key aspects of the case.

---

**strategy**

**Core Objective**: To precisely verify key facts and guide clients in providing admissible evidence, thereby ensuring clarity regarding the core elements of the case and reinforcing the evidentiary chain.

**Key Points**:

1. **Restricting the Scope of Answers**: Questions should be concrete and specific to ensure that the client's responses focus on crucial facts. Such questions often require "yes/no" answers or specific factual details, such as time, location, or object attributes.

2. **Efficient Information Verification**: This strategy is used to verify significant facts in the case, eliminate irrelevant information, and quickly narrow down the investigative focus or identify key evidentiary elements.

3. **Guiding Evidence Provision**: Lawyers should direct clients to potential sources or formats of evidence, ensuring that verbal statements are converted into verifiable forms. This includes asking about mediums such as WeChat messages, phone calls, and whether recordings or photographs exist.

4. **Managing Conversational Focus**: When clients deviate from the topic, closed-ended questions can be used to refocus the dialogue on the core of the case, thereby ensuring accuracy and completeness in evidence collection.

**Examples**:

- "Was the defendant holding a black umbrella in their left hand at the time?"

- "Was the contract signed on May 10, 2023?"

- "You mentioned the other party made threats---through what channel? Could you provide a call log or recording?"

---

**Strategy 3: Corroborative Detail Verification.** This strategy involves cross-validating information from multiple perspectives to ensure completeness and internal consistency. This strategy focuses on follow-up questions that clarify vague points, recover missing details, or identify contradictions. It examines facts through varied lenses, including individual behavior, environmental context, and supporting evidence.

---

**strategy**

**Core Objective**: To thoroughly investigate details, cross-validate facts, and encourage clients to reflect on the case from different perspectives, thereby ensuring the completeness and reliability of the information and identifying potential inconsistencies.

**Key Points**:

1. **Probing for Critical Details**: For vague or missing elements in the client's narrative, follow-up questions should be posed regarding specific times, locations,

---

```
        individual actions, and dialogue to collect more
        comprehensive details.

     2. Cross-Dimensional Fact Validation: Facts should
        be verified from multiple dimensions---such as the
        behavior of parties involved, environmental conditions,
        and sources of evidence---to ensure logical consistency
        and uncover potential contradictions.

     3. Introducing Hypotheticals and Supplementary Reflection:
        Hypothetical scenarios can be used to guide the client
        in considering alternative outcomes, which may lead to
        the discovery of overlooked or unexpressed details.

Examples:

     • "Can you describe in detail the suspect's clothing and
       physical characteristics at the time?"

     • "Please elaborate on the environmental
       conditions---such as lighting and background
       noise---during the incident."

     • "You mentioned it was pouring rain on the day of
       signing. Do you recall what you wore and what means of
       transportation you used? Could weather records serve
       as corroboration?"

     • "If you had made a different choice at that moment, how
       do you think the situation would have unfolded?"
```

**Strategy 4: Confrontational Disclosure Questioning.** This strategy targets information that the user may deliberately withhold due to its sensitive or unfavorable nature. This strategy prompts users to elaborate on omitted details by drawing attention to gaps in their narrative.

```
strategy

Background:
This strategy is based on the observation that clients
may selectively withhold unfavorable information unless
explicitly asked. Specifically:

     • When asked directly and precisely about adverse
       information, clients tend to answer truthfully.

     • When not directly questioned, clients may conceal or
       partially disclose such information.

Core Objective:
To identify areas where the client may be withholding
information and use targeted questioning to compel the
disclosure of unfavorable facts, ensuring that the case is
based on a full and accurate account.

Key Points:

     1. Direct Targeting of Concealed Information:
        By hypothesizing and refining questions, lawyers can
        zero in on potentially concealed key facts. The
        goal is to design questions that make it difficult
        for clients to avoid or partially answer. Emphasis
        should be placed on specific factual details (e.g.,
        time, place, individuals, transaction contents) that
        may force clients to admit previously undisclosed
        information.
```

```
   2. Exposing Known Inconsistencies:
      Drawing on existing evidence or known facts, lawyers
      can pose reverse questions based on inconsistencies to
      increase pressure on clients.  Comparing the client's
      account with other known information or evidence can
      prompt them to clarify or admit hidden aspects.
Examples:
      • "You mentioned in your previous statement that
        only you and the other party were present at the
        scene.  However, surveillance footage shows that
        additional individuals were nearby.  Could you provide
        their contact details or explain why they were not
        mentioned?"
      • "In your statement, you referred to a 'cooperation
        matter' discussed during the meeting, but did not
        elaborate on the specifics.  We obtained meeting notes
        that referenced 'financial terms' and 'additional
        commitments'.  Could you explain this content and
        clarify why it was not included in your statement?"
```

**LLM-Strategy: Adaptive Generative Questioning.** This strategy adopts a dynamic policy-generation mechanism. Prior to each conversational round, the system generates a lawyer-specific policy for the subsequent round of questioning with a large language model, based on the complete cumulative dialogue history of the case and a predefined set of prompts. This policy-generation process is recursive: as the dialogue progresses, the model continually performs conditional generation over the evolving dialogue state and the strategic prompts, iteratively producing new lawyer policies until the prescribed maximum number of turns for the case is reached.

```
strategy

Background:  To generate effective questioning strategies
for lawyers when communicating with clients, ensuring a
comprehensive and accurate understanding of case facts,
evidence, and the client's position, thereby laying the
groundwork for subsequent legal services and case strategy
formulation.
Role:  You are an expert in legal conversation strategy
generation, well-versed in various questioning techniques
used in lawyer{client communications.  Based on the dialogue
content, you are able to generate efficient and professional
next-step questioning strategies that enable the lawyer
to obtain more comprehensive information.  During the
communication process, clients may intentionally conceal
information unfavorable to them.
Task:  Based on the dialogue record between the lawyer and
the client, generate the next-step questioning strategy
that the lawyer may adopt, with the goal of maximizing the
retrieval of comprehensive information.
Dialogue History:  {history}
Output Requirements (strict structured format):
Core Objective:  A concise description of the central aim of
the current round of questioning.
Key Points:  A bullet-point list of strategies or techniques
that should be emphasized in this round of questioning.
Examples:  Provide 1{2 specific question formulations that
```

```
could be applied in this round.
Example:
``
Core Objective:  To precisely confirm key facts and guide
the client to provide verifiable information, ensuring that
the central facts of the case are clear while simultaneously
strengthening the chain of evidence.
Key Points:
    • Restrict the scope of answers:  Questions should
      be specific and precise, ensuring that the client's
      responses focus exclusively on the key facts, typically
      yielding "yes/no" or concrete detail answers (e.g.,
      time, place, object attributes).
    • Efficient fact confirmation:  Used to verify critical
      facts within the case, exclude irrelevant information,
      and rapidly narrow the investigative scope or lock in
      essential evidentiary details.
    • Guide evidence provision:  Direct the client to
      disclose possible sources or forms of evidence,
      ensuring that verbal statements are converted into
      verifiable forms.  This includes inquiring about
      carriers of evidence (e.g., WeChat, telephone) and
      whether recordings or photographs exist.
    • Steer the dialogue pace:  When the client digresses
      from the main topic, employ closed-ended questions to
      maintain focus on the case core, thereby ensuring both
      the accuracy and completeness of evidence collection.
Examples:
    • "Was the defendant holding a black umbrella in his left
      hand at that time?"
    • "Was the contract signed on May 10, 2023?"
    • "You mentioned that the other party threatened you;
      through what channel did this occur?  Can you provide
      the call record?"
''
```

## D  DATASET CONSTRUCTION PROCEDURES

To support our study on adaptive information elicitation in legal dialogue scenarios, we construct a structured dataset named *IELegal*, derived and refined from raw data in the CAIL2018 corpus (Xiao et al., 2018). All data are anonymized to ensure the privacy of individuals involved. The dataset construction consists of three main stages including factual content processing, key information extraction, and information categorization.

**Factual Content Processing.**  The original `fact` field in CAIL2018 often contains both objective case descriptions and subjective judicial conclusions, such as sentencing or legal qualifications. As our task focuses on eliciting user responses grounded in pre-trial facts, we aim to isolate the pure narrative of events. Given the unstructured nature of the fact field, rule-based filtering is insufficient. To address this, we employ a large language model (LLM) to preprocess each fact description, removing legal conclusions and retaining only the objective event details.

**Key Information Extraction.**  We then apply the LLM to extract structured key factual elements from the cleaned case narratives. The extracted elements include: (1) *time* of the incident, (2)

*location*, (3) *persons involved* (e.g., suspect, victim, or witnesses), and (4) *event details* (e.g., actions, consequences, and objects used). These structured facts serve as the reference ground truth for evaluating the effectiveness of factual information elicitation during multi-turn dialogue.

**Information Categorization.**  To simulate realistic user behavior under varying motivations, we further categorize each extracted event based on its favorability toward the defendant. Each element is labeled as:

- **Positive**: e.g., evidence of cooperation, absence of criminal intent.
- **Negative**: e.g., admission of violence, signs of premeditation.
- **Neutral**: background or legally irrelevant details.

This classification supports the development of a user simulation module in which the simulated user may selectively withhold or partially disclose unfavorable information. This behavior introduces a realistic challenge to the information elicitation process by mimicking the strategic tendencies observed in real-world legal consultations.

**IELegal-augment Augmentation.**  To address the variance in factual density across legal cases, where some entries contain rich contextual details while others are relatively sparse, we construct an augmented version of the dataset, referred to as *IELegal-augment*. This augmentation is performed using a large language model that is prompted to expand under-specified cases by generating plausible and contextually appropriate factual elaborations. The augmented content is constrained to remain consistent with the original charges and verified facts of each case.

**Dataset statistics.**  The dataset used in our experiments consists of a total of 2,000 legal case samples, divided into two subsets: *IELegal-base* and *IELegal-augment*, each containing 1,000 samples. For both subsets, the dataset are evenly split into training and testing sets, with 500 samples in each, resulting in a 0.5 training-to-testing ratio. Each sample is accompanied by structured factual and categorical fields to support downstream dialogue simulation and evaluation. Each data entry includes the following components:

- **Case_Content**: Basic case narrative information extracted from judicial documents.
- **Case_Type**: The type of legal case (e.g., theft, fraud, assault).
- **Plaintiff** and **Defendant**: The key parties involved in the case.
- **Key_Information**: Structured factual elements extracted by the LLM, including time, location, participants, and event descriptions.
- **Classified_Information**: Aggregated case events labeled by factual polarity (positive, negative, neutral).
- **Negative_Information**: Subset of case facts potentially adverse to the defendant.
- **Positive_Information**: Subset of case facts favorable to the defendant, such as cooperation or extenuating circumstances.
- **Neutral_Information**: Legally irrelevant or descriptive information not clearly favoring either side.

D.1   DATASET PROMPT DESIGNS

**Information Extraction Prompt.**  This prompt is designed to extract factual elements from a legal case description. It emphasizes the removal of formal or evaluative language such as official narratives or legal commentary, in order to ensure a clear and objective extraction of relevant factual content.

> **Example Case**
>
> **Task:**  Given a legal case description, extract only the
> factual components of the case while eliminating any formal

```
or evaluative language (e.g., official narratives or legal
commentary).
Case Description:
Output Requirements:
    • Only return the distilled factual content related to
      the case.
    • Remove any official, rhetorical, or non-factual
      statements.
    • Do not include any additional explanatory text or
      commentary in the output.
```

**Key Information Prompt.** This prompt is designed to extract essential information such as time, location, individuals involved, and events from unstructured factual descriptions of legal cases.

**Example Case**

```
Task:  Given the following factual description of a legal
case, organize the information into a structured format.
Input:
Output Format:
    {
      "Time": "",
      "Location": "",
      "People": "",
      "Events": [
        "1. ...",
        "2. ..."
      ]
    }
Requirements:
    • Break down the factual sequence into discrete,
      logically coherent events.
    • Ensure that each event item maintains internal
      consistency and reflects the progression of facts.
    • Output should strictly follow the specified JSON format
      without additional commentary or interpretation.
```

**Classified Information Prompt.** This prompt is designed to guide the model in categorizing case-related events as positive, neutral, or negative from the perspective of the defendant.

**Example Case**

```
Task:  You are an experienced judge reviewing a legal case.
Based on the information provided below, please assume the
perspective of the defendant, and classify the case details
into three distinct categories:
    1. Negative Information:  Elements that may increase
       {criminal}'s legal exposure or aggravate potential
       penalties.
    2. Positive Information:  Factors that may enhance
       {criminal}'s likelihood of acquittal or successful
       defense.
```

```
        3. Neutral Information:  Facts that do not significantly
           influence the outcome of the case or the legal position
           of the parties involved.
   Case Description:
   Requirements:
        • Perform the classification strictly based on the facts
          mentioned above.
        • If certain categories have no relevant information,
          indicate with "None".
        • Return the structured output in the following JSON
          format:
          {
            "Positive Information": [],
            "Negative Information": [],
            "Neutral Information": []
          }
```

**Extended Information Prompt.** This prompt is designed to encourage the model to expand the original case by generating additional contextual details. These may include an extended timeline of events, hypothetical evidence, plausible witness testimonies, legal arguments, relevant precedents, background information about the defendant, and potential implications of the case outcome.

---
**Example Case**

```
Role:  You are an expert in reconstructing legal case
narratives.
Task:  Given a partial description of a legal case, your
task is to enrich the case narrative by inferring and
supplementing plausible missing details, thereby producing
a more comprehensive and coherent case account.
Input:  {}
Notice:
    • Based on the provided partial case information, infer
      and add appropriate details to form a complete and
      logically consistent case narrative.
    • The output should be a single, continuous paragraph
      that reads as a coherent and realistic legal case
      description.
    • Avoid providing fragmented additions or segmented
      supplements|generate a fully integrated case narrative
      instead.
    • Maintain a formal and objective tone consistent with
      legal and academic standards.
```
---

## D.2 DATA EXAMPLE

**Initial Case Example.** We present an example to illustrate the structure of the original dataset.

**Example Case**

```
1   {
2     "fact": "At approximately 22:00 on May 16, 2014, the victim,
          Zhou Moumou, was driving a car near the entrance of a
          residential complex in Longgang District when he was
          stopped by the defendant, Lu Mou. A dispute arose over the
           unpaid referral fee from Zhou Moumou owed to Lu Mou.
          During the altercation, Lu punched Zhou in the face and
          nose, causing injuries later identified as minor.",
3     "meta": {
4       "relevant_articles": [234],
5       "accusation": ["Intentional Injury"],
6       "criminals": ["Liu Mou"],
7       "term_of_imprisonment": {
8         "death_penalty": false,
9         "imprisonment": 12,
10        "life_imprisonment": false
11      }
12    }
13  }
```

**Key Information Example.**   This example illustrates the key information extracted from the original case data.

**Example Case**

```
1   {
2     "time": "22:00, May 16, 2014",
3     "location": "Entrance of a residential garden, Longgang
          District",
4     "persons": {"victim": "Zhou Moumou","defendant": "Lu Mou"},
5     "Events": [
6       "The victim, Zhou Moumou, was driving a car through the
            entrance of a residential garden in Longgang District.",
7       "The defendant, Lu Mou, intercepted Zhou Moumou.",
8       "A dispute arose due to the unpaid referral fee of Zhou
            Moumou owed to Lu Mou.",
9       "Lu Mou assaulted Zhou Moumou by punching him in the face
            and nose.",
10      "The injuries sustained by Zhou Moumou were classified as
            minor injuries of the second degree."
11    ]
12  }
```

**Classified Information Example.**   This example presents the extracted case information organized according to predefined classification categories.

**Example Case**

```
1   {
2     "Negative Information": [
3       "Lu Mou inflicted injuries on Zhou Moumou by punching him in
            the face and nose.",
4       "The injuries sustained by Zhou Moumou were medically
            classified as minor injuries of the second degree."
5     ],
6     "Positive Information": [
```

```
 7      "The dispute arose because Zhou Moumou owed Lu Mou a
            referral fee, which may be relevant to understanding the
            motivation of Lu Mou and the surrounding context.",
 8      "The incident occurred at night, which may affect the
            reliability of eyewitness testimony and the collection
            of evidence."
 9    ],
10    "Neutral Information": [
11      "The incident occurred at 22:00 on May 16, 2014.",
12      "The location of the incident was the entrance of a
            residential garden in Longgang District.",
13      "The victim was Zhou Moumou, and the defendant was Lu Mou."
14    ]
15  }
```

