# OpenReview forum: "RPS: Information Elicitation with Reinforcement Prompt Selection"
_ICLR.cc/2026/Conference — Submitted to ICLR 2026_

### Official Review · Reviewer_kz24 · 2025-10-23

**Soundness:** 2
**Presentation:** 2
**Contribution:** 2
**Rating:** 2
**Confidence:** 3

**Summary:**

This paper tackles the problem of enabling large language models (LLMs) to adaptively elicit user-known but concealed information in open-ended conversations — a key challenge in domains like law, healthcare, and counseling where users may withhold sensitive facts. The authors propose Reinforcement Prompt Selection (RPS), a lightweight reinforcement learning framework that formulates prompt selection as a sequential decision-making task. Instead of relying on static or handcrafted prompts, RPS learns a policy over a pool of prompt strategies to maximize information gain through dialogue, using a normalized reward function that stabilizes learning. The method is first validated in a synthetic Gaussian Mixture Model environment that simulates user disclosure bias, and then tested on IELegal, a newly introduced benchmark built from real legal case records designed to mimic realistic, biased user behavior. Experimental results show that RPS consistently outperforms static and LLM-generated prompting strategies, demonstrating its effectiveness in adaptively uncovering relevant and sensitive information. The work provides a principled foundation for interactive AI systems that can engage in information-seeking dialogues with socially aware and context-sensitive adaptability.

**Strengths:**

1. The paper explicitly defines the Information Elicitation (IE) problem — how to extract information users know but hesitate to share — and positions it as a fundamental limitation of current LLM-based dialogue systems. It introduces a socially grounded motivation, bridging LLM dialogue modeling with human disclosure behavior (positive, neutral, negative information). This gives conceptual clarity and interdisciplinary depth rarely seen in prompt-optimization work.
2. Reinforcement Prompt Selection (RPS) is a lightweight yet principled RL framework that models prompt choice as a sequential decision problem, rather than as static prompt engineering or token-level optimization. The use of normalized information gain as a reward is technically sound — it stabilizes training and avoids reward vanishing as the dialogue progresses. The framework is domain-agnostic, modular (separable from the base LLM), and computationally efficient compared to token-wise RLPrompt-style methods.
3. The authors provide both controlled synthetic validation (via a Gaussian Mixture Model with disclosure bias) and realistic domain evaluation (the new IELegal dataset). The dual evaluation design (synthetic + real-world) effectively demonstrates the method’s adaptability and robustness to both unbiased and biased disclosure settings. Quantitative metrics (KL divergence, semantic similarity via Sentence-BERT) are clearly justified and aligned with the objective of information recovery.
4. The IELegal dataset is an important contribution: a legally grounded benchmark derived from real criminal cases, simulating realistic user reluctance and factual complexity. It provides structured factual annotations, bias labeling (positive/neutral/negative), and multi-turn dialogue simulations, offering the community a concrete platform to study information elicitation in sensitive domains.
5. The work aligns well with practical dialogue systems in law, healthcare, and education — where adaptive questioning is essential for completeness and trustworthiness. The ethical statement is careful: experiments use only anonymized, public data; there’s no deployment risk.

**Weaknesses:**

1. While the application to information elicitation is new, the underlying reinforcement-based prompt optimization mechanism is conceptually similar to prior works such as RLPrompt (Deng et al., 2022) and TEMPERA (Zhang et al., 2022). The main technical step — selecting discrete prompt templates via an RL policy — is incremental rather than fundamentally new, relying on established DQN/DDPG setups without introducing algorithmic innovation. The novelty thus primarily lies in **problem framing and dataset construction**, not in RL or LLM adaptation mechanics.
2. Both evaluation setups (GMM environment and IELegal benchmark) depend on synthetic or scripted user behavior, where user disclosure follows pre-defined rules (positive/neutral/negative bias). These simulations lack real psychological variability — users in real conversations may conceal information for nuanced emotional, cultural, or contextual reasons. This gap makes it unclear whether the model would generalize to true human dialogues, where “concealment” is not linearly related to sentiment polarity.
3. The normalized information gain reward presupposes that an accurate “ground-truth information set” $I$ exists and that semantic distance $d(\hat{I}_{t}, I)$ can be computed. In practical LLM dialogue, true user information is inaccessible, and similarity-based proxies (e.g., Sentence-BERT embeddings) can misjudge factual correctness or relevance. Hence, the **reward design is not deployable in real-world systems**, limiting practical applicability to simulation-only settings.
4. The baselines are primarily rule-based or handcrafted prompt strategies plus one LLM-generated adaptive method. Missing comparisons include stronger meta-prompting or self-play baselines, such as: (1) LLMs fine-tuned via _preference optimization_ (RLHF, DPO, PPO-style methods); (2) Active questioning models (e.g., Socratic prompting or iterative retrieval-augmented agents). Without these, the performance gap may be overstated relative to more capable contemporary methods.
5. While the authors acknowledge ethical risks, the system’s objective — to elicit concealed user information — has potentially intrusive implications. There is no detailed discussion of safeguards, consent mechanisms, or human oversight, which would be essential in sensitive applications (law, health, personal dialogue).

**Questions:**

1. The reward depends on a distance $d( \hat{I}\_{t}, I )$, but what distance metric ensures semantic and factual fidelity? How sensitive is the learning outcome to the choice of Sentence-BERT similarity? Is the normalization (division by the previous distance) numerically stable when $d( \hat{I}_{t-1}, I)$ becomes small?
2. How large is the prompt pool, and how are the prompts generated or curated? Are they manually designed or learned automatically? How sensitive is the policy performance to the number or diversity of prompt templates?
3. Why were only handcrafted prompt strategies and one LLM-generated policy included as baselines?
4. The RL reward requires access to ground-truth $I$. How could this be replaced or approximated in real applications?

**Details Of Ethics Concerns:**

The system’s objective to elicit concealed user information has potentially intrusive implications.

---

> ### Author Response · Authors · 2025-11-24
> **Response(1/3)**
>
> We sincerely appreciate your constructive feedback. We address your specific concerns below.
>
> **Response to Weakness 1**
>
> >"While the application to information elicitation is new, the underlying reinforcement-based prompt optimization mechanism is..."
>
> In addition to the novelty of our problem formulation and dataset, our method introduces meaningful methodological differences from prior reinforcement learning approaches to prompt optimization. Prior work primarily focuses on generating or editing prompts by finetuning or backpropagating through the entire backbone LLM. In contrast, our approach uses a predefined set of domain-specific prompts and trains an RL policy to select among them according to the evolving dialogue context. This design greatly reduces the computational cost of reinforcement learning. Moreover, because information elicitation requires a stable and domain-agnostic reward signal, we utilize a pretrained Sentence-BERT model for similarity matching, which provides consistent and reliable feedback. As a result, the overall framework remains lightweight, efficient, and readily adaptable to different data domains.
>
> **Response to Weakness 2**
>
> >"Both evaluation setups (GMM environment and IELegal benchmark) depend on synthetic or scripted user behavior..."
>
> We acknowledge that differences exist between simulated dialogues and interactions with real users. However, our dialogue data are collected from real legal cases and then de-identified, which helps narrow this gap. It is common in real scenarios for individuals to avoid disclosing information that may be disadvantageous to themselves, especially in legal consulting. Based on our interviews with lawyers and practicing legal professionals, their feedback confirms that this behavior occurs frequently. To reflect this tendency, we also instruct the LLM to withhold true information that may be unfavorable to the simulated user. This design choice allows the setting to more closely mirror actual conversational dynamics. Conducting studies with real users is an important direction that we plan to pursue in future work.
>
> **Response to Weakness 3**
>
> >"The normalized information gain reward presupposes that an accurate "ground-truth information set" $\mathcal{I}$ exists and that semantic distance $d(\hat{\mathcal{I}}_t, \mathcal{I})$ can be computed..."
>
> We respectfully disagree that our reward design limits practical applicability. The reward function is required only during training to guide policy learning and is not used during deployment. During training, we extract ground-truth information from de-identified legal cases to compute the reward. Once training is complete, the system relies solely on the learned policy selector and does not require ground-truth information or reward computation. Moreover, the reward model is a pretrained and domain-agnostic Sentence-BERT encoder that needs no additional training, which makes the training process lightweight and broadly applicable. Because reward computation is fully decoupled from deployment, our framework remains suitable for real-world use.
>
> **Response to Weakness 4**
>
> >"The baselines are primarily rule-based or handcrafted prompt strategies plus one LLM-generated adaptive method. Missing comparisons include..."
>
> Our approach both reduces the computational cost of prompt optimization and achieves strong performance in information elicitation. In contrast to prior work that primarily focuses on generating or editing prompts by finetuning or backpropagating through the entire backbone LLM, our approach uses a predefined set of domain-specific prompts and trains an RL policy to select among them based on the evolving dialogue context.
>
> For active questioning baselines, our Adaptive Generative Questioning strategy already represents an LLM-driven questioning approach aligned with the methods you mentioned, and our experiments show that it remains less effective than our proposed framework. We have also incorporated experiments with GRIPS. GRIPS performs rule-based prompt editing through simple operations on the instruction, but it is not well suited for leveraging domain knowledge in legal consulting. Specifically, we use each of the first four domain knowledge based strategies as the initial prompt for GRIPS. After each dialogue turn, GRIPS dynamically adjusts the strategy based on the dialogue history (for example, by deleting, swapping, or adding segments). The updated prompt is then used for the next turn. As shown in Figure 4, even with these GRIPS-based modifications, its performance remains lower than that of our proposed method and sometimes performs worse than simply using domain-specific prompts. Additional details are provided in Appendix B.2.1.

---

> ### Author Response · Authors · 2025-11-24
> **Response(2/3)**
>
> **Response to Weakness 5**
>
> >"While the authors acknowledge ethical risks, the system’s objective — to elicit concealed user information — has potentially intrusive implications. There is..."
>
> The primary goal of this work is to assist users in addressing the challenges they face and to reduce negative outcomes that may arise from incomplete or unclear disclosures. For example, in legal consulting, clients may omit or downplay important facts, which can limit a lawyer’s ability to prepare an effective defense and ultimately lead to unfavorable consequences for the client. In this context, the intended use of our framework is to support users rather than to exploit or mislead them. We have discussed the potential risks associated with eliciting concealed information in the ethical statement section of the paper and outlined the corresponding safeguards. We also agree that measures such as transparency, informed consent, and appropriate human oversight are essential for deployment in sensitive domains, and we view these considerations as important directions for future work.
>
> **Response to Question 1**
>
> >"The reward depends on a distance $d(\hat{\mathcal{I}}_t, \mathcal{I})$, but what distance metric ensures semantic and factual fidelity? How sensitive is the learning outcome..."
>
> In the LLM-based dialogue experiments, we provide an explicit definition of the distance metric: $d(\hat{\mathcal{I}}_t, \mathcal{I})$ is computed as the cosine similarity between Sentence-BERT embeddings. This metric quantifies the degree of semantic alignment between the extracted information and the ground-truth facts. We also acknowledge that the choice of Sentence-BERT model may affect the learning outcomes, since it directly shapes the reliability of the reward signal used during training. Regarding normalization, we consider it essential for ensuring stable reward scaling across dialogue turns. Without normalization, two issues arise:
>
> **• Reward signal instability and early-stage bias:** Without normalization, information gain tends to be large in the early stages of a dialogue, because matching a substantial portion of the ground-truth information is relatively easy at the beginning. As the dialogue progresses, the remaining information becomes increasingly limited, causing the reward to naturally diminish and approach zero. This results in high reward signals in the early turns and negligible signals in the later ones. Such imbalance can lead to gradient vanishing or learning stagnation, and it biases the agent toward exploiting early “large gains” while lacking sufficient incentive to continue refining its questioning strategy and eliciting the remaining hidden information once it is close to the ground truth.
>
> **• Training instability and low efficiency:** Without normalization, the reward distribution exhibits high variance due to the non-stationary nature of dialogue-based reinforcement learning. As reward scales fluctuate across different dialogue states, the policy becomes overly sensitive to stochastic spikes, leading to unstable or oscillatory updates and slower convergence. Reward normalization mitigates this variance, stabilizes learning dynamics, and prevents the training process from stalling as the agent approaches an effective policy.
>
> **Response to Question 2**
>
> >"How large is the prompt pool, and how are the prompts generated or curated? Are they manually designed or learned automatically? How sensitive is the policy performance to the number or diversity of prompt templates?"
>
> Our strategy prompt pool consists of five strategies, including four handcrafted strategies and one LLM-generated policy. Each handcrafted strategy prompt is approximately 500 characters in length, and these prompts are distilled from established legal literature and reference materials. We manually design these strategy prompts to demonstrate that our framework is flexible in incorporating domain knowledge. In particular, we can create prompts tailored to a specific field, such as law or mental health counseling, ensuring that the resulting strategies align more closely with professional questioning practices in that domain. Regarding the number and diversity of strategy prompts in the pool, these factors reflect the amount of domain expertise encoded in the system. Intuitively, a larger and more diverse prompt pool provides broader domain coverage and can therefore lead to better overall performance.

---

> ### Author Response · Authors · 2025-11-24
> **Response(3/3)**
>
> **Response to Question 3**
>
> >"Why were only handcrafted prompt strategies and one LLM-generated policy included as baselines?"
>
> We have now incorporated GRIPS as a new baseline. GRIPS performs rule-based prompt editing through simple operations on the instruction, but it is not well suited for leveraging domain knowledge in legal consulting. Specifically, we use each of the first four domain knowledge based strategies as the initial prompt for GRIPS. After each dialogue turn, GRIPS dynamically adjusts the strategy based on the dialogue history (for example, by deleting, swapping, or adding segments). The updated prompt is then used for the next turn. As shown in Figure 4, even with these GRIPS-based modifications, its performance remains lower than that of our proposed method and sometimes performs worse than simply using domain-specific prompts. Additional details are provided in Appendix B.2.1.
>
> **Response to Question 4**
>
> >"The RL reward requires access to ground-truth $ \mathcal{I} $. How could this be replaced or approximated in real applications?"
>
> In practical applications, the reward function is required only during training to guide policy learning and is not used during deployment. The ground-truth information $ \mathcal{I} $ are extracted from training data, e.g. real legal cases. Furthermore, the reward model is a pretrained, domain-agnostic Sentence-BERT encoder that requires no additional fine-tuning, making the training process lightweight and widely applicable. Because reward computation is fully decoupled from the deployment phase, the proposed framework remains feasible and effective for real-world use.

---

### Official Review · Reviewer_P4sL · 2025-10-30

**Soundness:** 3
**Presentation:** 3
**Contribution:** 3
**Rating:** 6
**Confidence:** 3

**Summary:**

This paper addresses a critical challenge for Large Language Models (LLMs) in open-ended dialogues: effectively eliciting information that users know but may intentionally conceal (e.g., sensitive information in legal or medical scenarios). It proposes a lightweight reinforcement learning framework called Reinforcement Prompt Selection (RPS). The core contribution is modeling the Information Elicitation (IE) problem as a sequential decision-making task. Instead of generating prompts token-by-token, the RPS framework learns a policy network that adaptively selects the optimal prompting strategy from a predefined pool (containing strategies like exploratory or adversarial prompts). This selection is based on the current dialogue history and acquired information, aiming to maximize information gain in subsequent interactions. To validate the method, the paper conducts two types of experiments: 1. ​​Synthetic Experiments: In a controlled Gaussian Mixture Model (GMM) environment, it demonstrates that an RL agent using RPS outperforms random baselines in information recovery accuracy (measured by KL divergence). 2. ​​Real-world Scenario Experiments: It constructs a new legal domain benchmark dataset,  IELegal(with IELegal-base and IELegal-augment subsets), based on real legal cases that simulate a user (defendant) concealing unfavorable facts. Results show that RPS significantly outperforms various static prompt baselines and an "adaptive generative questioning" baseline in extracting key facts from simulated users.

**Strengths:**

1. The paper accurately identifies a core pain point for LLMs in high-stakes applications (e.g., law, healthcare): users intentionally concealing information due to privacy, social desirability, or conflicting interests. Framing this as the IE problem and incorporating the social psychology concept of information valence (positive, neutral, negative) is highly relevant and forward-thinking.

2. The RPS framework strikes an excellent balance between efficiency and effectiveness. Compared to token-level prompt generation methods like RLPrompt, selecting from a predefined prompt pool significantly reduces computational overhead and deployment difficulty, making it more suitable for real-time dialogue systems.

​​3. The IELegal dataset is a significant contribution. Based on real cases, with structured fact annotations and simulated user disclosure biases, it provides a much-needed benchmark for the sub-field of "eliciting concealed information."

**Weaknesses:**

1. Both the GMM environment and the LLM/rule-based user simulation in IELegal (e.g., "omit or partially conceal unfavorable information if the question is not direct or precise") are far from the complex psychology of real humans (e.g., changing trust, emotional fluctuations, active deception). The lack of evaluation with real human users (Human-in-the-Loop) casts doubt on RPS's real-world effectiveness.

​​2. The IELegal dataset is relatively small (1000 cases per subset) and confined to the legal domain. The medical and financial scenarios mentioned in the introduction are not validated, leaving the cross-domain generalization capability of RPS (and the cost of migrating the prompt pool) unclear.

​​3. The "adaptive generative questioning" baseline is relatively weak (it lets an LLM generate a strategy rather than execute one). A fairer and stronger comparison would be against state-of-the-art LLM agent frameworks (e.g., ReAct, Self-Ask) or a powerful LLM granted the same prompt pool (as in-context examples) and instructed to "select the best strategy and generate a question."

**Questions:**

1. The core of RPS relies on a predefined prompt pool, constructed for IELegal based on domain knowledge (legal interview techniques). If migrating RPS to a new domain (e.g., medical consultation), what would be the manual cost of building this pool? Are there plans for automated or semi-automated prompt pool construction?

2. ​​In a real scenario where a user is determined to conceal information (e.g., outright denying a key fact in a legal consultation), RPS would receive near-zero rewards consistently. Could this lead to policy collapse or getting stuck in ineffective questioning loops?

3. ​​Do the authors plan to (or have they conducted preliminary) test RPS with real human users to validate its effectiveness beyond simulated environments?

---

> ### Author Response · Authors · 2025-11-24
> **Response**
>
> Thanks for the reviewer's comments. Here are our responses to the comments.
>
> **Response to Weakness 1  and Question 3**
>
> >"W1: Both the GMM environment and the LLM/rule-based user simulation..."
> >
> >"Q3: Do the authors plan to..."
>
> We acknowledge that differences exist between simulated dialogues and interactions with real users. However, our dialogue data are collected from real legal cases and then de-identified, which helps narrow this gap. It is common in real scenarios for individuals to avoid disclosing information that may be disadvantageous to themselves, especially in legal consulting. Based on our interviews with lawyers and practicing legal professionals, their feedback confirms that this behavior occurs frequently. To reflect this tendency, we also instruct the LLM to withhold true information that may be unfavorable to the simulated user. This design choice allows the setting to more closely mirror actual conversational dynamics. Conducting studies with real users is an important direction that we plan to pursue in future work.
>
> **Response to Weakness 2 and Question 1**
>
> >"W2: The IELegal dataset is relatively small..."
> >
> >"Q1: The core of RPS relies on a predefined prompt pool, constructed..."
>
> Regarding dataset scale and domain coverage, we plan to incorporate additional domain-specific datasets in the future, such as for mental health counseling, to broaden our empirical evaluation.
>
> The current prompt pool is indeed insufficient for fully general-purpose information elicitation. However, the goal of our design is not to construct a universal prompt set applicable to all tasks. Instead, the framework is built to support the creation of domain-specific prompt pools. By incorporating specialized knowledge from a given domain, we can develop prompts tailored to that domain. This design is inherently flexible and allows the framework to be readily adapted to practical, domain-specific settings. In addition, our use of the pretrained Sentence-BERT requires no finetuning, because the similarity matching reward is naturally domain-agnostic and remains compatible with different prompt pools. Therefore, instead of developing generic automatic prompt generation, we focus on expanding additional domain-specific datasets in the future.
>
> **Response to Weakness 3**
>
> >"The "adaptive generative questioning" baseline is relatively weak (it lets an LLM generate a strategy rather than execute one). A fairer and stronger comparison would be..."
>
> We would like to clarify that the LLM-generated strategy was designed to operate without explicit domain knowledge, rather than to select an optimal strategy from a shared prompt set. Empirically, this generative baseline underperforms several handcrafted strategies, indicating that LLM-only prompting is less reliable for consistently eliciting key factual elements. This further highlights the importance of domain-grounded strategies and adaptive selection.
>
> We have now incorporated GRIPS as a new baseline. GRIPS performs rule-based prompt editing through simple operations on the instruction, but it is not well suited for leveraging domain knowledge in legal consulting. Specifically, we use each of the first four domain knowledge based strategies as the initial prompt for GRIPS. After each dialogue turn, GRIPS dynamically adjusts the strategy based on the dialogue history (for example, by deleting, swapping, or adding segments). The updated prompt is then used for the next turn. As shown in Figure4, even with these GRIPS-based modifications, its performance remains lower than that of our proposed method and sometimes performs worse than simply using domain-specific prompts. Additional details are provided in Appendix B.2.1.
>
> **Response to Question 2**
>
> >"In a real scenario where a user is determined to conceal information (e.g., outright denying a key fact in a legal consultation), RPS would receive..."
>
> Although we acknowledge that extreme scenarios can exist in real-world interactions, these are outside the scope of the present work. We address this concern from two perspectives:
>
> **• Training.** The reward function is used only during the reinforcement learning process, and the training data are derived from real-world legal cases. Because these cases provide complete factual information, we are able to extract the true underlying details needed for reward computation. Consequently, extreme scenarios in which a user deliberately withholds all information do not appear in the training data.
>
> **• Deployment.** Our framework assumes that the user is genuinely seeking assistance rather than engaging in deliberately adversarial behavior, as might occur in a criminal interrogation, which is not the target scenario of our application. In our setting, withholding information is treated as unconscious or unintentional rather than a purposeful attempt to deceive. Therefore, we do not expect such extreme adversarial situations to occur in our target application.

---

### Official Review · Reviewer_b7ep · 2025-11-04

**Soundness:** 3
**Presentation:** 3
**Contribution:** 2
**Rating:** 4
**Confidence:** 3

**Summary:**

The paper presents RPS, which provides a framework for building LLMs capable of eliciting concealed information via dialogues. The key to the method is learning a policy over a pool of different prompts that determines the next query for the LLM to ask. The policy is trained over the reward of maximizing information gain. The authors evaluate RPS on synthetic dialogue, and a real-world legal case benchmark.

**Strengths:**

1. The paper formalizes the problem of information elicitation, which is an important and practical problem.

2.  The approach is lightweight by training a policy over a finite set of prompts, rather than open-ended token generation. This avoids having to train the LLM itself, which reduces computational overhead.

3. The evaluation on legal cases is interesting, and can be used broadly as a benchmark for open-ended dialogue.

**Weaknesses:**

1. The biggest weakness lies in its limited evaluation. Specifically, the evaluation is limited to synthetic and legal case dialogues. This limits the generalizability of the findings for all open-ended dialogues requiring information elicitation. Furthermore, in both domains the user is simulated using a simple prompt to discourage revelation of information. It is likely this does not model the complexity of dynamics of a real-world conversations; specifically, the simulated users in the evaluation have no temporal dynamics.

2. The performance is measured based on similarity of sentence embeddings. It is unclear if such reward truly captures the nuances of how good the policy did, compared to human or even LLM-based evaluations of the full dialogue.

3. The authors compare against a variety of prompting baselines. While the baselines do consider a variety of strategies, and even adapting between them, the comparison feels unfair as RPS requires additional training and additional overhead during inference. The results would be stronger if the authors also compared to a training baseline, or even more sophisticated inference such as GRIPS that the authors mentioned in related work.

**Questions:**

1. Currently, RPS considers a concise but potentially restrictive action space of predefined prompts. Do the authors believe the current set of prompts is sufficient for general information elicitation? If not, how would one expand the action space to fit to more complex tasks?

2. The authors introduce a normalized reward signal based on information gain. What happens if the unnormalized information gain was used as the reward?

---

> ### Author Response · Authors · 2025-11-24
> **Response(1/2)**
>
> We sincerely appreciate your constructive feedback. We address your specific concerns below.
>
> **Response to Weakness 1**
>
> >"The biggest weakness lies in its limited evaluation. Specifically, the evaluation is limited to..."
>
> Although our current evaluation covers only a limited set of scenarios, the framework has demonstrated its effectiveness within these settings. We are actively extending our evaluation to additional domains, including mental health counseling, to further validate its generality. Regarding the absence of temporal dynamics in the real-world conversations, we plan to conduct studies involving real users in future work to address this limitation.
>
> **Response to Weakness 2**
>
> >"The performance is measured based on similarity of sentence embeddings. It is unclear if such reward truly captures..."
>
> We argue that a reward based on text similarity is better suited to our setting than either human evaluation or LLM-as-a-judge, for three main reasons:
>
> **1. Task-specific suitability.**
>
> Our goal is to recover the complete set of underlying ground-truth information. Sentence BERT has been shown to correlate strongly with sentence-level semantic similarity, making it a natural choice for computing rewards based on domain-agnostic similarity matching. In addition, by leveraging the multi-turn structure of the dialogue, we design a reward computation method that operates along the dimensions of the ground-truth information units. This method aggregates evidence across turns to evaluate the extent to which the true information has been elicited.
>
> **2. Limitations of LLM-as-a-judge.**
>
> Large language models are prone to hallucinations, and there is no universally accepted, fine-grained evaluation standard for our specific task. As a result, LLM-as-a-judge can produce unstable and inconsistent assessments across repeated evaluations of the same dialogue. In addition, invoking a large model incurs significantly higher computational and monetary costs compared with computing sentence-level similarity using Sentence-BERT.
>
> **3. Limitations of human evaluation.**
>
> Human evaluation can be incorporated in two ways: humans acting as users and humans serving as reward annotators. Both directions are promising and will be explored in future work. However, purely human-based evaluation presents several challenges. Evaluation criteria are difficult to standardize not only across different annotators, but also for the same annotator over time, which can lead to inconsistent and unstable scoring. In addition, human evaluation is extremely time-consuming and labor-intensive, resulting in substantially higher costs compared with our Sentence-BERT based reward mechanism.
>
> **Response to Weakness 3**
>
> >"The authors compare against a variety of prompting baselines. While the baselines do consider a variety of strategies, and even..."
>
> We fully acknowledge your suggestion and have revised the paper by adding experiments involving GRIPS. GRIPS performs rule-based prompt editing through simple operations on the instruction, but it is not well suited for leveraging domain knowledge in legal consulting. Specifically, we use each of the first four domain knowledge based strategies as the initial prompt for GRIPS. After each dialogue turn, GRIPS dynamically adjusts the strategy based on the dialogue history (for example, by deleting, swapping, or adding segments). The updated prompt is then used for the next turn. As shown in Figure 4, even with these GRIPS-based modifications, its performance remains lower than that of our proposed method and sometimes performs worse than simply using domain-specific prompts. Additional details are provided in Appendix B.2.1.
>
> **Response to Question 1**
>
> >"Currently, RPS considers a concise but potentially restrictive action space of predefined prompts. Do the authors believe..."
>
> The current prompt pool is indeed insufficient for fully general-purpose information elicitation. However, the goal of our design is not to construct a universal prompt set applicable to all tasks. Instead, the framework is built to support the creation of domain-specific prompt pools. By incorporating specialized knowledge from a given domain, we can develop prompts tailored to that domain. This design is inherently flexible and allows the framework to be readily adapted to practical, domain-specific settings. In addition, our use of the pretrained Sentence-BERT requires no finetuning, because the similarity matching reward is naturally domain-agnostic and remains compatible with different prompt pools.

---

> ### Author Response · Authors · 2025-11-24
> **Response(2/2)**
>
> **Response to Question 2**
>
> >"The authors introduce a normalized reward signal based on information gain. What happens if the unnormalized information gain was used as the reward?"
>
> Using unnormalized information gain as a reward may lead to the following issues:
>
> **• Reward signal instability and early-stage bias:** Without normalization, information gain tends to be large in the early stages of a dialogue, because matching a substantial portion of the ground-truth information is relatively easy at the beginning. As the dialogue progresses, the remaining information becomes increasingly limited, causing the reward to naturally diminish and approach zero. This results in high reward signals in the early turns and negligible signals in the later ones. Such imbalance can lead to gradient vanishing or learning stagnation, and it biases the agent toward exploiting early “large gains” while lacking sufficient incentive to continue refining its questioning strategy and eliciting the remaining hidden information once it is close to the ground truth.
>
> **• Training instability and low efficiency:** Without normalization, the reward distribution exhibits high variance due to the non-stationary nature of dialogue-based reinforcement learning. As reward scales fluctuate across different dialogue states, the policy becomes overly sensitive to stochastic spikes, leading to unstable or oscillatory updates and slower convergence. Reward normalization mitigates this variance, stabilizes learning dynamics, and prevents the training process from stalling as the agent approaches an effective policy.

---

### Official Review · Reviewer_TpMQ · 2025-11-05

**Soundness:** 3
**Presentation:** 3
**Contribution:** 3
**Rating:** 6
**Confidence:** 4

**Summary:**

The paper addresses the challenge of eliciting "user-known but concealed information" in open-ended dialogues with Large Language Models (LLMs). Authors state this is an important problem in sensitive domains like legal or clinical support, where users may withhold information due to privacy concerns or social hesitation. (I am not sure about this statement though)

The authors propose Reinforcement Prompt Selection (RPS), a lightweight RL framework. Instead of generating prompts token-by-token (which is computationally expensive), RPS learns a policy (using DQN) to select the best prompt from a predefined pool of strategies (e.g., "Exploratory," "Confrontational") at each turn of the conversation. The goal is to maximize the amount of information recovered from the user.

To evaluate RPS, the authors introduce a new benchmark dataset, IELegal, constructed from real-world, anonymized legal cases. They test their method in both a synthetic Gaussian Mixture Model (GMM) environment and on the IELegal benchmark (using an LLM to simulate a biased user). The results show that RPS outperforms static, fixed-prompting baselines in recovering concealed information.

**Strengths:**

- The paper clearly formulates the information elicitation problem. As LLMs become integrated into sensitive applications, the ability to adaptively elicit relevant information may be helpful in certain scenarios (like legal).

- The core idea of selecting from a prompt pool rather than generating prompts is a practical simplification. This makes the RL problem more tractable, reduces computational overhead, and is a sensible design choice for real-world deployment. However,

- The creation and public release of the IELegal dataset is valuable (from real legal documents).

- The work sits at the intersection of reinforcement learning, sequential decision-making, and large language models. These are very active areas of research for the ICLR community.

**Weaknesses:**

- The core idea of selecting from a prompt pool rather than generating prompts is a practical simplification. This makes the RL problem more tractable, reduces computational overhead, and is a sensible design choice for real-world deployment. However, if computational overhead is not a constraint, one can utilize the selected prompt (or relevant information) to craft/generate a better prompt. This would make the real system more usable.
- the empirical evaluation is solid but still somewhat small (1k-case) legal dataset, and all “users” simulated by LLMs. Also, it is unclear how English prompts with Chinese legal cases confound the LLM's results. More datasets would be useful to consider to add more weight to the research analysis.
- the work doesn’t justify the choice of algorithms (DQN, etc) vs more modern or simpler alternatives (contextual bandits, supervised classifier over prompts, PPO, GRPO). The paper mentions RLprompt work in related work, but doesn't compare to this method either (or GRIPS); it may not be suitable to compare this work to RLPrompt, etc; the baselines used in this paper look reasonable but clearly not SOTA

**Questions:**

- Why did you not consider comparing RPS to other RL-based prompt optimization methods like RLPrompt, even if adapted to a selection framework? A PPO-based agent on the same discrete action space would be a more direct and stronger baseline.
- This research's motivation is to "uncover information users may intentionally withhold." Could you elaborate on the safeguards necessary to prevent this from being "weaponized" as a tool for social engineering? How do you distinguish between ethical "elicitation" and manipulative "extraction"?

**Details Of Ethics Concerns:**

- I don't really think ethics review is needed here, however, since the paper is on the topic of information elicitation, it will be good for responsible AI reviewers to take a quick look at this work.

---

> ### Author Response · Authors · 2025-11-24
> **Response**
>
> Thanks for the reviewer's comments. Here are our responses to the comments.
>
> **Response to Weakness 1, Weakness 3 and Question 1**
>
> >"W1: The core idea of selecting from a prompt pool..."
> >
> >"W3: the work doesn’t justify the choice of algorithms..."
> >
> >"Q1: Why did you not consider comparing RPS to..."
>
> We would like to clarify that comparing different RL algorithms (e.g., PPO vs. DQN) is not the primary focus of this paper. Our core contribution lies in the novel formulation of the problem and the framework. Specifically, our framework introduces a reinforcement learning (RL) based mechanism that selects the next action from a prompt pool, choosing the questioning strategy most likely to elicit hidden information given the current dialogue state.
>
> We have now incorporated GRIPS as a new baseline. GRIPS performs rule-based prompt editing through simple operations on the instruction, but it is not well suited for leveraging domain knowledge in legal consulting. Specifically, we use each of the first four domain knowledge based strategies as the initial prompt for GRIPS. After each dialogue turn, GRIPS dynamically adjusts the strategy based on the dialogue history (for example, by deleting, swapping, or adding segments). The updated prompt is then used for the next turn. In contrast, our framework is built to support the creation of domain-specific prompt pools. By incorporating specialized knowledge from a given domain, we can develop prompts tailored to that domain. This design is inherently flexible and allows the framework to be readily adapted to practical, domain-specific settings. In addition, our use of the pretrained Sentence-BERT requires no finetuning, because the similarity matching reward is naturally domain-agnostic and remains compatible with different prompt pools. As shown in Figure 4, even with these GRIPS-based modifications, its performance remains lower than that of our proposed method and sometimes performs worse than simply using domain-specific prompts. Additional details are provided in Appendix B.2.1.
>
> **Response to Weakness 2**
>
> >"the empirical evaluation is solid but still somewhat small (1k-case) legal dataset, and all "users" simulated by LLMs. Also, it is unclear how English prompts..."
>
> Regarding dataset scale and domain coverage, we plan to incorporate additional domain-specific datasets in the future, such as for mental health counseling, to broaden our empirical evaluation.
>
> **Response to Question 2**
>
> >"This research's motivation is to "uncover information users may intentionally withhold." Could you elaborate on the safeguards necessary to prevent..."
>
> The primary goal of this work is to assist users in addressing the challenges they face and to reduce negative outcomes that may arise from incomplete or unclear disclosures. For example, in legal consulting, clients may omit or downplay important facts, which can limit a lawyer’s ability to prepare an effective defense and ultimately lead to unfavorable consequences for the client. In this context, the intended use of our framework is to support users rather than to exploit or mislead them. We have discussed the potential risks associated with eliciting concealed information in the ethical statement section of the paper and outlined the corresponding safeguards. We also agree that measures such as transparency, informed consent, and appropriate human oversight are essential for deployment in sensitive domains, and we view these considerations as important directions for future work.

---

### Author Response · Authors · 2025-11-24
**General Response**

We sincerely thank all reviewers for their detailed reviews and invaluable feedback. These constructive comments have been crucial in identifying areas for improvement and enhancing the overall quality of our work.

Before addressing each specific concern, we would like to clarify the core contribution of our paper. Our framework introduces a reinforcement learning (RL) based mechanism that selects the next action from a prompt pool, *choosing the questioning strategy most likely to elicit hidden information given the current dialogue state*. Our approach both reduces the computational cost of prompt optimization and achieves strong performance in information elicitation. In contrast to prior work, which primarily focuses on generating or editing prompts by finetuning or backpropagating the entire backbone LLM, our approach uses a predefined set of domain-specific prompts and trains an RL policy to select among them based on the evolving dialogue context. Furthermore, due to the unique nature of information elicitation, we utilize a pretrained Sentence-BERT model for domain-agnostic similarity matching, which provides stable and consistent reward signals for RL. As a result, the overall framework remains lightweight, efficient, and easily adaptable to different data domains.

We have incorporated GRIPS and RLPrompt as new baselines. For GRIPS, it performs rule-based prompt editing via simple operations on the instruction. We adapt the original GRIPS framework to the multi-turn dialogue setting as follows: We use each of the first four domain knowledge based strategies as an initial prompt for GRIPS. After each dialogue turn, GRIPS dynamically adjusts the strategy based on the dialogue history (e.g., by deleting, swapping, or adding segments), and the updated prompt is then used in the next turn. For RLPrompt, it introduces a gradient-free RL framework that optimizes discrete prompt generation through sequential token selection, obviating the need for manual design. We adopt the complete RLPrompt framework for discrete prompt generation. To maintain a fair comparison, we substitute its original reward function with the one proposed in this paper. As shown in Figure 4, the results show that RPS consistently achieves the highest overall performance. Neither GRIPS nor RLPrompt surpasses RPS under any setting, and in most cases their performance is only comparable to that of the original fixed strategies. These findings suggest that rule-based prompt editing, whether based on simple instruction modifications as in GRIPS or reinforcement-based adjustments as in RLPrompt, has limited effectiveness in capturing and leveraging domain-specific knowledge in legal consultation and potentially in other specialized domains. Further implementation details are provided in Appendix B.2.1 and B.2.2.

---

### Meta-Review · Area_Chair_pWhh · 2026-01-05

**Summary:**

The paper presents Reinforcement Prompt Selection (RPS), a lightweight framework that uses a Deep Q-Network (DQN) to select questioning strategies from a predefined pool to elicit concealed information from users. While the reviewers appreciated the introduction of the IELegal dataset and the clear formulation of the "Information Elicitation" (IE) problem—particularly its relevance to high-stakes domains like law—there is a consensus that the work remains somewhat preliminary for a top-tier venue. The primary concerns center on the methodological novelty (viewed by several reviewers as an incremental application of standard RL to a discrete prompt-selection task) and the narrowness of the evaluation. The reliance on LLM-simulated users and synthetic Gaussian Mixture Models (GMM) was a recurring point of contention, as these simulations likely fail to capture the psychological complexity, emotional nuances, and varied deception strategies of real human participants in sensitive contexts.

**Reviewer Concerns:**

While the authors provided a diligent rebuttal, several critical concerns remain outstanding. The authors successfully addressed the lack of competitive baselines by incorporating GRIPS and clarified that the ground-truth-dependent reward is strictly a training signal, not required at deployment. However, the most significant weakness—the lack of human-in-the-loop validation—remains unaddressed, with the authors deferring this to future work. Without evidence that a policy trained on simulated LLM defendants generalizes to real human behavior, the practical utility of RPS remains speculative. Furthermore, the concern regarding the simplicity of the RL approach (DQN over five static prompts) versus more modern, agentic reasoning frameworks (e.g., ReAct or Self-Ask) was not fully resolved, as the paper focuses on a relatively rigid action space that may not scale to the fluid nature of real-world dialogue.

**Reviewer Scores:**

Regarding the reviewer scores, the divergence suggests a split between those valuing the dataset/framing and those prioritizing methodological rigor. Reviewer TpMQ and Reviewer P4sL were initially positive due to the problem's importance; however, given the discussion on the "simulated user" gap, they likely would have trended toward a 5 (marginal reject) as the limitations of the evaluation became clearer. Reviewer b7ep remained skeptical of the evaluation's generalizability; even with the GRIPS baseline added, the absence of human data would likely prevent them from moving to an accept. Reviewer kz24 provided the most critical assessment, focusing on the incremental nature of the RL mechanism and the intrusive ethical implications.

While the authors clarified the deployment phase, the incremental critique is difficult to overcome without a more innovative algorithmic contribution, suggesting their score would likely remain a 2 or 3. Consequently, the lack of real-world validation and the incremental technical contribution lead to a recommendation of Rejection.

---

### Decision · Program_Chairs · 2026-01-26

Reject